# Goal Achievement Guided Exploitation:
# A Principled Performance-Based Scheduling Framework for Reinforcement Learning

**Shengchao Yan**[1]*     *yan@cs.uni-freiburg.de*

**Baohe Zhang**[1]*     *zhangb@cs.uni-freiburg.de*

**Chenguang Huang**[2]     *chenguang.huang.research@gmail.com*

**Joschka Boedecker**[1]     *jboedeck@cs.uni-freiburg.de*

**Wolfram Burgard**[2]     *wolfram.burgard@utn.de*

[1]*Department of Computer Science & IMBIT//BrainLinks-BrainTools, University of Freiburg*
[2]*Department of Computer Science and Artificial Intelligence, University of Technology Nuremberg*

**Reviewed on OpenReview:** *https://openreview.net/forum?id=uGidWOfKhK*

## Abstract

In dense-reward tasks, Reinforcement learning (RL) algorithms often employ soft entropy regularization to promote exploration. By integrating an entropy term into the objective function, they regularize exploration via tuning the coefficient. However, the entropy coefficient only indirectly influences the action distribution through gradient updates, making it difficult to precisely control exploration, and requires careful scheduling to balance exploration and exploitation throughout training. As a solution, we propose Goal Achievement Guided Exploitation (GAGE), a performance-based scheduling framework that adaptively regulates exploration by linking policy stochasticity directly to the agent's performance relative to a target value. Unlike soft entropy regularizers, GAGE enforces hard, performance-dependent constraints on action distribution's standard deviation for continuous actions and logit range for discrete actions. Consequently, GAGE ensures a guaranteed lower bound on action probabilities that naturally decays as the agent approaches optimal performance. Across a suite of challenging robotic control tasks, GAGE improves learning efficiency and stability across various strong baselines, achieving competitive or superior final performance. By providing a more interpretable and robust alternative to entropy-based exploration heuristics, GAGE offers a scalable path toward solving complex dense reward tasks with pronounced local optima.

## 1 Introduction

Local optima in the objective landscape pose a fundamental challenge for gradient-based optimization (Chaudhari et al., 2017), especially in high-dimensional robotic control tasks. Even in environments with dense rewards, the trial-and-error nature of reinforcement learning (RL) means agents often learn predominantly from suboptimal trajectories. As a result, RL algorithms are often highly sensitive to hyperparameters and prone to premature convergence to arbitrary local optima.

To mitigate this problem, entropy regularization has been widely integrated in RL algorithms. By augmenting the standard optimization objective with an entropy term that encourages policy stochasticity, entropy

---

*Equal contribution.

regularization promotes exploration and often yields more stable and robust policies (Williams & Peng, 1991; Mnih et al., 2016; Schulman et al., 2017; Haarnoja et al., 2018a). However, relying on a soft regularization through entropy coefficient adjustment also introduces practical limitations. It necessitates extensive hyperparameter tuning to manage the exploration-exploitation trade-off and sometimes requires a manual coefficient annealing schedule to ensure precise, low-entropy policies at convergence. The manual tuning of coefficient can be difficult to design and highly sensitive to reward scaling (Haarnoja et al., 2018a). Existing methods like Soft Actor-Critic (SAC-v2) (Haarnoja et al., 2018b) attempt to automate the scheduling by learning the entropy coefficient for the policy to match a fixed target entropy. However, this acts as a static soft constraint, which may be too high for fine-grained control or too low for initial exploration, failing to provide the adaptive decay required for convergence.

Despite various recent advances in exploration strategies (Ladosz et al., 2022; Wang et al., 2023; Wan et al., 2023; Yan et al., 2024b; Sukhija et al., 2025), the burden of tuning exploration schedules remains a barrier to scalable deployment. In this work, we propose Goal Achievement Guided Exploitation (GAGE), a framework that formalizes exploration annealing into a principled exploration scheduling strategy by leveraging the agent's *goal achievement*, defined as the ratio between its current performance and the optimal value. By utilizing this signal, our method adaptively reduces exploration as the agent approaches its desired performance, allowing the exploration rate to evolve naturally with learning progress rather than following a manually specified schedule. This is achieved by imposing a hard constraint on the action distribution through an adaptive constraint on policy stochasticity, implemented as a minimum standard deviation in continuous action spaces and a limited logit range in discrete ones.

We implement our approach across both continuous and discrete action spaces, as well as on- and off-policy algorithms. We evaluate GAGE on challenging locomotion and dynamic manipulation tasks involving quadruped and humanoid robots, along with benchmarks from HumanoidBench (Sferrazza et al., 2024), and show that our approach overall improves learning efficiency and performance compared to strong baselines. GAGE also demonstrates robustness to reward engineering. Furthermore, we demonstrate that the previously observed advantage of discrete over continuous policies in continuous control tasks (Tang & Agrawal, 2020) primarily arises from premature convergence, a limitation effectively alleviated by our method.

In summary, we make the following key contributions:

1. We propose **G**oal **A**chievement **G**uided **E**xploitation (GAGE), a novel framework that regulates exploration through an adaptive constraint linked to the agent's goal achievement, offering a principled performance-based scheduling framework that automates the exploration-exploitation trade-off.

2. We propose using hard constraint for different measures of distribution stochasticity, which are more interpretable and controllable than entropy-based soft regularization.

3. We integrate GAGE seamlessly with continuous and discrete action spaces, as well as on- and off-policy algorithms. We demonstrate the strong empirical performance of GAGE across challenging customized robot control tasks and the HumanoidBench benchmark.

4. We derive a theoretical lower bound on action probabilities guaranteed by our method, demonstrating its effectiveness with an empirical analysis.

## 2 Background and Related Work

Premature convergence is a long-standing challenge in optimization and machine learning, affecting algorithms including genetic algorithms (Pandey et al., 2014) and reinforcement learning Yan et al. (2024a). To clarify the specific focus of this work, we distinguish our focus on premature convergence from issues such as "reward hacking" and sparse rewards in RL, as elaborated in Appendix A.1. In the following, we review the principle of entropy regularization, analyze its limitations, and discuss related work.

## 2.1 Entropy Regularization in Reinforcement Learning

In the standard RL setting, we model the task as a Markov decision process defined by the tuple $(\mathcal{S}, \mathcal{A}, p, r)$, where the agent interacts with the environment following the policy $\pi(\cdot \mid s_t)$ with state $s_t \in \mathcal{S}$ and action $a_t \in \mathcal{A}$. In each time step, the environment transits from state $s_t$ to $s_{t+1}$ according to the state transition probability $p\colon \mathcal{S} \times \mathcal{S} \times \mathcal{A} \to [0, 1]$ and gives the agent a reward following the reward function $r\colon \mathcal{S} \times \mathcal{A} \to \mathbb{R}$. The objective of standard RL is to maximize the expectation of the cumulative reward $\sum_t \mathbb{E}_{(s_t, a_t) \sim \rho_\pi}[r(s_t, a_t)]$, where $\rho_\pi$ stands for the state-action distribution following policy $\pi$. The objective can be extended to infinite-horizon problems with a discount factor $\gamma$.

A common practice in RL is to employ an entropy regularization objective (Williams & Peng, 1991; Mnih et al., 2016; Schulman et al., 2017; Haarnoja et al., 2018a; Espeholt et al., 2018), which augments the standard optimization objective with the expected entropy of the policy, such as in SAC or PPO:

$$\sum_{t=0} \mathbb{E}_{(s_t, a_t) \sim \rho_\pi}[r(s_t, a_t) + \beta \mathcal{H}(\pi(\cdot \mid s_t))] \quad \text{or} \quad L_t^{\mathrm{CLIP}} - c L_t^{\mathrm{VF}}(\theta) + \beta \mathbb{E}_{s_t \sim \rho_\pi}[\mathcal{H}(\pi(\cdot \mid s_t))],$$

where $\mathcal{H}(\pi)$ denotes the policy entropy, $\beta, c$ are coefficients, and $L_t^{\mathrm{CLIP}}(\theta), L_t^{\mathrm{VF}}(\theta)$ are the clipped surrogate objective and value function loss. This formulation encourages exploration by promoting higher entropy.

**Manuel Exploration Scheduling**  The additional entropy term inevitably alters the optimization objective. While entropy maximization improves exploration and robustness (Ahmed et al., 2019), it can be detrimental in tasks that require a precise, low-entropy policy. To recover the conventional objective at convergence, the entropy temperature $\beta$ must be gradually reduced so that $\beta \to 0$. However, manuel schedule tuning can become prohibitively costly due to the high sensitivity of RL algorithms to hyperparameters. In contrast, our goal is to develop a more adaptive mechanism that automatically adjusts exploration without the need for such manual scheduling.

**Limited Interpretability and Controllability for Exploration**  The temperature $\beta$ is a sensitive hyperparameter that lacks a universal scale; its optimal value is coupled with the reward magnitude, action dimensionality, and the landscape of specific local optima (Haarnoja et al., 2018b). Rather than relying on soft regularization through temperature adjustment, we propose to impose hard constraints directly on the policy's action distribution to obtain more predictable and controllable exploration behavior.

## 2.2 Related Work

In this section, we review contemporary approaches to enhancing exploration in reinforcement learning.

**Adaptive hyperparameter tuning**  Soft Actor-Critic (SAC-v1) (Haarnoja et al., 2018a) is one of the most widely adopted off-policy RL algorithms using entropy-based regularization. However, its performance is highly sensitive to the temperature hyperparameter $\beta$, whose optimal value is non-trivial to tune. To address this issue, Haarnoja et al. (2018b) proposed learning a gradient-based $\beta$ that matches the expected entropy to a predefined target value. While this approach (SAC-v2) enables dynamic adjustment of $\beta$ during training, it shifts the tuning burden to the choice of the target entropy. The fixed entropy constraint also limits the policy's ability to either start learning with a higher exploration rate or converge to an optimal deterministic solution. To eliminate the need for additional hyperparameters, Wang & Ni (2020) applied a metagradient method (Xu et al., 2018) to tune $\beta$ automatically. However, the reported performance improvement over SAC is marginal, and metagradient updates can still become trapped in local optima due to their gradient-based nature.

**Intrinsic motivations**  While our work addresses premature convergence in dense-reward environments, another line of research focuses on encouraging exploration in sparse-reward tasks, where the optimization landscape is largely flat and external rewards are scarce. Inspired by the curiosity-driven behaviors of animals (Schmidhuber, 1991), intrinsic motivation methods encourage exploration by rewarding the agent for visiting novel or informative states. Count-based approaches (Bellemare et al., 2016; Tang et al., 2017)

achieve near-optimal exploration in tabular settings but scales poorly to high-dimensional or continuous domains. Prediction-error-based methods (Pathak et al., 2017; Burda et al., 2019) train forward or inverse dynamics models and use discrepancies between predicted and observed transitions as intrisic rewards. Zhang et al. (2021) proposed rewarding novelty differences between states to encourage breadth-first exploration, while Wan et al. (2023) scaled observation novelty using mutual information between states and trajectories. Although effective in sparse-reward environments such as MiniGrid (Chevalier-Boisvert et al., 2023) or navigation tasks, the applicability of these methods to high-dimentional continuous control task with severe local optima remains not clear. Moreover, balancing extrinsic and intrinsic rewards is non-trivial. MaxInfoRL (Sukhija et al., 2025) mitigates this issue by introducing information-based rewards and automatically tuning the temperature for the transition information gain. They reported superior results in off-policy continuous control tasks with humanoid robots. However, the method incurs significant computational overhead due to model ensembles and target policies, and it is limited to off-policy settings.

**Goal and trajectory planning**   The Markovian assumption in standard RL algorithms may be suboptimal for tasks involving partial observation or long-horizon dependencies. Recent works have approached exploration as a structured, long-term planning problem. Jain et al. (2023) utilized non-Markovian policies that condition on past trajectory to maximize state coverage within limited steps, achieving efficient exploration in gridworld and simple continuous control tasks such as Reacher and Pusher. Hu et al. (2023) leveraged learned world models and planning algorithms to generate exploratory goals with high exploration potential, effectively constructing a goal curriculum. Similarly, Diaz-Bone et al. (2025) quantify the achievability, novelty, and relevance of exploratory goals to guide exploration toward meaningful directions. However, these approaches generally rely on prior task knowledge and are primarily evaluated in low-dimensional action spaces or sparse-reward environments, such as navigation or gridworld tasks.

**Multi-modal policy**   In standard RL implementations such as SAC, the policy is typically modeled as a Gaussian distribution, whose unimodal nature limits the representation of multiple behavioral modes. Tang & Agrawal (2020) addressed this issue by discretizing continuous action spaces. With a sufficiently fine granularity (e.g., over 11 bins per action dimension), discrete policies outperformed Gaussian ones on most MuJuCo benchmarks. More recently, Dong et al. (2025) adopted diffusion models for policy representation, enabling flexible, multi-modal action distributions. They demonstrated marginal performance improvement on MuJuCo tasks. Despite their enhanced representation capacity, these approaches remain constrained by the same limitations inherent to entropy-based exploration discussed earlier. In this work, we provide implementations using both Gaussian and discrete policies. We identify combining diffusion-based policies with our framework as a promising direction for future research.

## 3   Goal Achievement Guided Exploitation

To account for the limitations of entropy regularization, we propose a novel framework named Goal Achievement Guided Exploitation (GAGE). In the optimal case, a policy should not converge before the agent approaches its maximum achievable performance. Therefore, it is natural to relate the policy's convergence level to its goal achievement. In this section, we first define goal achievement formally, then describe how we use it to construct an adaptive constraint for policy convergence, and finally analyze the benefits of GAGE.

### 3.1   Goal Achievement

Since an agent is trained to maximize the cumulative reward, we define its goal achievement as

$$g(\pi) \coloneqq \frac{J_\pi}{J_{\pi^*}}, \qquad J_\pi = \sum_{t=0}^{T} \mathbb{E}_{(s_t, a_t) \sim \rho_\pi}[r(s_t, a_t)]. \tag{1}$$

By definition, $g(\pi) \leq 1$ and $g(\pi) \to 1$ as $\pi \to \pi^*$. The expected return of the current policy $J_\pi$ can be estimated using the average return of recent rollouts. However, determining the optimal expected return $J_{\pi^*}$ is sometimes difficult for complex reward functions. In addition, some reward components may not fully align with the task goal, potentially leading to increased $g(\pi)$ without achieving the actual goal.

To address these issues, we examine the structure of the reward function more closely. A typical reward function consists of multiple terms, which can be categorized into *goal rewards* ($r_{\text{goal}}$) and *auxiliary rewards* ($r_{\text{aux}}$). Goal rewards are mandatory and correspond to the intended task objective, such as winning a game or executing a robotic behavior. Auxiliary rewards are optional heuristics designed to guide or accelerate learning. We exclude auxiliary rewards when measuring goal achievement for two reasons: (1) they may not align with the true task goal and can lead to suboptimal solutions; (2) their maximum attainable values are often difficult to define, whereas the maximum goal reward values are typically available in the task specification. Accordingly, we define the goal achievement for the goal reward as:

$$g(\pi) := \frac{J_{\text{goal},\pi}}{J_{\text{goal},\pi^*}}, \qquad J_{\text{goal},\pi} = \sum_{t=0}^{T} \mathbb{E}_{(s_t,a_t) \sim \rho_\pi}[r_{\text{goal}}(s_t, a_t)].$$

In practice, we approximate goal achievement using the moving average of the per-episode ratio between the cumulative goal reward $\sum_{t=0}^{T} r_{\text{goal},t}$ and its corresponding maximum value $\sum_{t=0}^{T} r_{\text{goal},t}^{\max}$.

Tasks can contain multiple goal reward terms (Xu et al., 2020; Hayes et al., 2022). In this work, we focus on single-goal settings and leave the multi-objective extension for future research. We primarily consider non-negative rewards. For negative rewards, transformations such as sigmoid or offset can ensure $g(\pi) \in [0, 1]$. If individual components are unavailable, $g(\pi)$ can be approximated using cumulative return of total rewards as Eq. 1. When the maximum return is unknown, the optimal performance can be estimated empirically from observed trajectories, as further demonstrated in Sec. 4.1.

## 3.2 Mitigating Premature Convergence via Action Smoothing

To prevent policy collapse, where the agent prematurely converges to a few actions in discrete spaces or an excessively narrow Gaussian in continuous spaces, we introduce an *action smoothing* technique. It avoids overconfidence by lower-bounding the action space coverage of the policy distribution according to the current goal achievement. Concretely, for each action space type, we select a scalar quantity $\mathcal{F}(\pi)$ that captures action space coverage. We refer to this informally as the *flatness* of the policy.

We then enforce an adaptive lower bound $\mathcal{F}_{\text{LB}}$ on the flatness, requiring $\mathcal{F}(\pi) \geq \mathcal{F}_{\text{LB}}$ throughout training. This lower bound is a decreasing function of goal achievement:

$$\mathcal{F}_{\text{LB}}(\pi) := f(g(\pi)).$$

As a simple heuristic, we choose $f$ to be an affine function of the goal achievement, as specified in Eqs. 2 and 5, leaving investigation of alternative mappings for future work. We next specify $\mathcal{F}(\pi)$ and its implementation for continuous and discrete action spaces. Full algorithmic details with different backbone algorithms are outlined in Appendix B.

**Continuous action space** In continuous domains, exploration is typically facilitated by modeling actions as Gaussian distributions: $p(a \mid s) \sim \mathcal{N}(\mu(s), \sigma^2)$. This formulation is used in both stochastic policies, such as SAC and Proximal Policy Optimization (PPO) (Schulman et al., 2017; Haarnoja et al., 2018a), and deterministic ones such as Deep Deterministic Policy Gradient (DDPG) (Lillicrap et al., 2016), where Gaussian noise is added for exploration. We employ the standard deviation $\sigma$ of the policy as the flatness measure since larger $\sigma$ directly corresponds to broader action space coverage. The corresponding lower bound $\sigma_{LB}$ (i.e., $\mathcal{F}_{\text{LB}}$ for the continuous case) is heuristically set as:

$$\sigma_{\text{LB}}(\pi) = -\sigma_0 g(\pi) + \sigma_0, \tag{2}$$

where the hyperparameter $\sigma_0 > 0$ controls the minimum allowed standard deviation when goal achievement $g(\pi)$ is zero. Action smoothing is applied by clamping $\sigma$ to $\sigma_{\text{LB}}$, ensuring $\sigma \geq \sigma_{\text{LB}}$. When $\sigma_0 = 0$, the formulation reduces to the original baseline algorithms. Although entropy contains similar information to standard deviation for Gaussian distributions, directly constraining $\sigma$ is more straightforward.

**Discrete action space**   For discrete actions $a_k$, where $k \in \{1, \ldots, K\}$, probabilities are computed as:

$$p(a_k \mid s) = \text{softmax}(z)_k = \frac{\exp(z_k)}{\sum_{i=1}^{K} \exp(z_i)}, \tag{3}$$

where $z = (z_1, \ldots, z_K) \in \mathbb{R}^K$ are the logits output from the policy network. Analogous to $\sigma$ for Gaussian distribution in continuous spaces, the range of logits reflects the flatness of the categorical distribution. We thus use the *negative logit range* $\delta_z$ as $\mathcal{F}(\pi)$ to ensure that all actions retain a minimum probability:

$$\delta_z := -\max_{i,j=1,\ldots,K} |z_i - z_j| = \min_k z_k - \max_k z_k, \tag{4}$$

and following the same heuristic as in continuous space, the corresponding lower bound $\delta_{z,\text{LB}}$ (i.e., $\mathcal{F}_{\text{LB}}$ for the discrete case) is set as:

$$\delta_{z,\text{LB}} = (\delta_{z,1} - \delta_{z,0})g(\pi) + \delta_{z,0}, \tag{5}$$

where $\delta_{z,0}$ and $\delta_{z,1}$ are hyperparameters controlling the minimum allowed $\delta_z$ at $g(\pi) = 0$ and $g(\pi) = 1$. In principle, $\delta_{z,1} \to -\infty$ as the policy approaches deterministic. However, this would make it numerically difficult to calculate Eq. 5. In practice, we find $\delta_{z,1} \approx -15$ is sufficient. We implement this constraint using *softmax with adaptive temperature* (Asadi & Littman, 2017), $\text{softmax}(z/\tau)$, which controls the distribution flatness by adaptively scaling logits before exponentiation. The adjusted logits are computed as:

$$z' = \frac{z}{\tau}, \quad \tau = \frac{\max\{1, |\delta_z|\}}{|\delta_{z,\text{LB}}|}, $$

after which action probabilities are calculated via $\text{softmax}(z')$. The temperature $\tau$ rescales the logits so that $\delta_{z'} \geq \delta_{z,\text{LB}}$. In practice, enforcing a flat distribution with a hard constraint on the softmax logits alone can result in prohibitively large logit magnitudes. To address this issue, we regularize the logits by minimizing:

$$\mathcal{L}_z := \alpha \sum_{t=0}^{T} \mathbb{E}_{s_t \sim \rho_\pi} \|z(s_t)\|_2^2, \tag{6}$$

where $\alpha$ is a regularization coefficient. It's worth noting that label smoothing (Szegedy et al., 2016), another technique for distribution flattening, is not suitable for action smoothing. We explain our choice for softmax with temperature in Appendix A.2 in detail.

## 3.3   Advantages of GAGE

**Automated exploration scheduling**   GAGE enables a dynamic exploration rate without manuel parameter scheduling, making it easier to tune and generalize. Furthermore, since the constraint depends solely on goal achievement, GAGE promotes more stable learning dynamics that are less sensitive to local optima.

**Analytical guarantees for non-vanishing exploration**   GAGE imposes hard lower bounds on the flatness of the action distribution, ensuring that exploration never vanishes. This formulation analytically lower-bounds the probability of all actions for both continuous and discrete spaces. For Gaussian policy, this property follows naturally from the lower-bounded standard deviation. For discrete actions, the probability bound after action smoothing can be derived explicitly from Eqs. 3 and 4:

$$\min_k p(a_k \mid s) = \frac{\exp(\min_k z_k)}{\sum_{i=1}^{K} \exp z_i} \geq \frac{\exp(\max_k(z_k) + \delta_{z,\text{LB}})}{\exp(\max_k(z_k) + \delta_{z,\text{LB}}) + (K-1)\exp(\max_k z_k)} = \frac{\exp \delta_{z,\text{LB}}}{\exp \delta_{z,\text{LB}} + (K-1)}.$$

Thus, the minimum action probability is adaptively lower-bounded as a function of the goal achievement $g(\pi)$. This mechanism prevents the irreversible loss of potentially optimal behaviors as shown in Fig. 7.

## 4   Experiments

This section validates GAGE on robot control problems characterized by high degrees of freedom and local optima, often leading to premature convergence. We show (i) how GAGE achieves superior on- and off-policy exploration efficacy compared to baselines, (ii) how GAGE demonstrates strong robustness to unknown optimal goals and variations in reward shaping, and (iii) how GAGE generalizes to discretized action spaces.

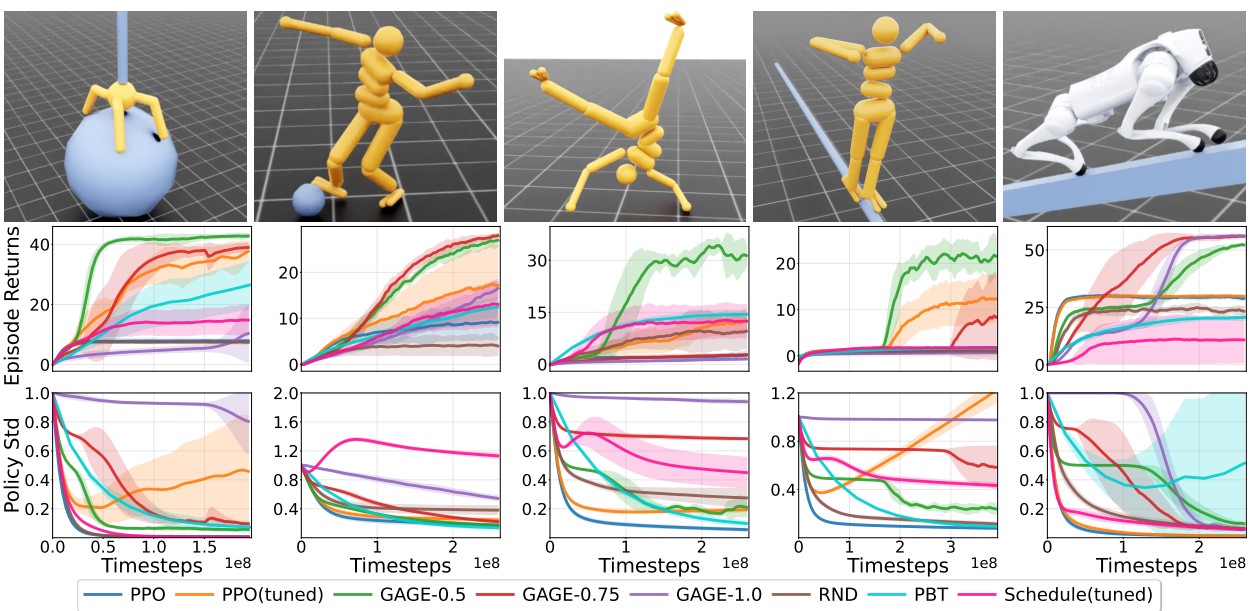

Figure 1: Mean episode returns and action standard deviation $\sigma$ over 5 seeds; shaded regions denote one standard deviation. We denote GAGE-0.5 as using $\sigma_0 = 0.5$. **Top**: tasks (left to right), Ant Acrobatics, Humanoid Dribbling, Humanoid Cartwheel, Humanoid Tightrope, and Dog (Unitree Go2) Balance Beam. **Middle**: training curves of episode returns. **Bottom**: Averaged policy standard deviation $\bar{\sigma}$.

## 4.1 GAGE for Continuous Action Space

We integrate GAGE into both on-policy and off-policy algorithms by replacing entropy regularization with our adaptive constraint. First, we combine GAGE with PPO, which is widely used to learn challenging robot control tasks due to its benefits from large-scale GPU parallelism (Makoviychuk et al., 2021). To test GAGE under severe local optima issues, we design five highly challenging tasks in IsaacLab (Mittal et al., 2023; Yan et al., 2024b) (see Fig. 1). The environment details are provided in Appendix D.1. We then incorporate GAGE into SAC-v2 and evaluate SAC-GAGE, on over 10 HumanoidBench (Sferrazza et al., 2024) tasks.

## On-Policy RL with GAGE

We compare PPO-GAGE against: (i) vanilla PPO with default hyperparameters, (ii) PPO with per-task tuned hyperparameters, (iii) Random Network Distillation (RND) (Burda et al., 2019), an intrinsic motivated method that encourages exploration of novel states and is commonly used in sparse-reward settings; (iv) Population-Based Training (PBT) (Henderson et al., 2018), which trains a population of PPO agents with different entropy coefficients in parallel and adapts these coefficients online based on episode return performance (details in Appendix C); and (v) hand-designed entropy-coefficient schedules (Appendix C). Among the schedule variants we evaluated, cosine decay performed best and is therefore included as our representative scheduling baseline. We include RND to test whether the curiosity-driven method can mitigate premature convergence in dense-reward settings. To obtain strong baselines, we perform a grid search over the PPO exploration hyperparameter, the entropy coefficient, per task and adopt the best-performing values (see Appendix D.2 for details). For RND We follow the hyperparameter settings of the original and subsequent work of Yang et al. (2024). PPO-GAGE sets entropy temperature $\beta = 0$ and uses the same hyperparameters as vanilla PPO for a direct comparison. With action spaces normalized to $[-1, 1]$, intuitively, we set $\sigma_0 \in \{1.0, 0.75, 0.5\}$ for an ablation study.

Fig. 1 shows learning curves for episode returns and average $\sigma$ across robot joints. PPO-GAGE with $\sigma_0 = 0.5$ successfully solved all the tasks, whereas vanilla PPO failed. To ensure a fair comparison with the baselines, in this work, we only report the total episode return including both the goal reward and auxiliary rewards. Hyperparameter optimisation (HPO) over the entropy coeffcient substantially improves PPO on

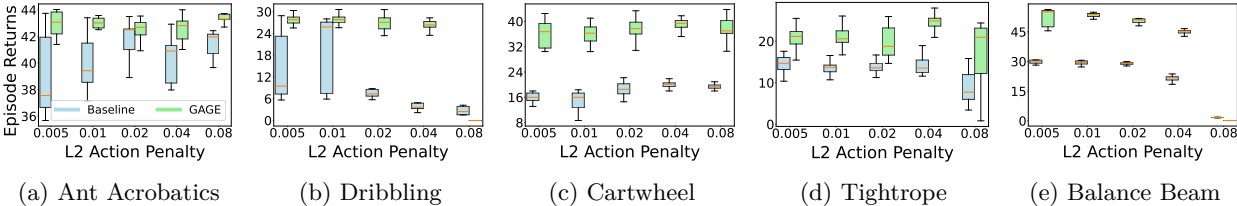

Figure 2: Box plots of episode returns (excluding action-penalty values) for tuned PPO versus PPO-GAGE. The $x$-axis shows the investigated coefficients of the squared L2 action penalty $\|a\|_2^2$. We aggregate the last 10 episode returns of each seed for 50 data points per box.

Ant Acrobatics, Humanoid Dribbling, Humanoid Cartwheel, and Humanoid Tightrope. However, on the Balance Beam task, HPO yields no noticeable improvement. Among all tasks, PPO-GAGE still wins by clear margins. PPO-GAGE with $\sigma_0 = 0.5$ or $0.75$ performs reliably across the majority of tasks. In contrast, agents with $\sigma_0 = 1.0$ exhibit poorer performance, as the excessively large action variance produces highly random behaviors. Hence, for practical use, we recommend starting with $\sigma_0$ values between 0.5 and 0.75.

The $\sigma$ plots reveal that vanilla PPO quickly reduces policies' standard deviation at the start of training, achieving higher rewards by over-exploiting reward components such as energy cost or survival. PPO continues to decrease $\sigma$ even after the target rewards plateau. For instance, the dog robot learns to stabilize on the beam and ceases exploring despite a forward motion target. In contrast, PPO-GAGE sustains exploration and only reduces $\sigma$ when the desired behaviors are learned (see Appendix D.1).

In both the Ant Acrobatics and Humanoid Tightrope environments, we observe that optimally tuned PPO agents exhibit a divergent increase in the policy parameter $\sigma$ following an initial decay, even as task performance reaches a plateau. This phenomenon suggests that the agent exploits the entropy regularization term by injecting excessive variance into task-irrelevant action dimensions, specifically, joints non-critical to maintaining the local optima. These empirical findings underscore a significant objective mismatch: the pressure to maximize entropy drives the policy toward high-variance regimes that are functionally redundant, potentially destabilizing the learning process.

RND successfully slows down the reduction of $\sigma$ relative to PPO. However, the additional exploration does not solve the tasks and can even hurt performance. We hypothesize that RND's novelty-seeking can be counterproductive in high-dimensional control where most "novel" states are failed trajectories (e.g., falling down). This result highlights the different focuses between premature convergence and the sparse reward problem. Further experimental results with varying intrinsic reward settings are provided in Fig. 11.

Notably, the best fixed entropy coefficient (tuned PPO results) outperforms both PBT-based entropy tuning and the best-performing hand-designed entropy schedule. This suggests that, in our dense-reward control tasks, frequently changing the entropy coefficient during training can be detrimental, potentially because it induces a non-stationary regularization and thus a moving optimization target.

**Improved Robustness to Reward Shaping**

Reward shaping is crucial yet delicate, as small changes to reward weights can result in unsuccessful learning. We suppose the sensitivity to reward shaping largely arises from premature convergence and conduct experiments to test the robustness of our method to variations in reward shaping.

The L2 norm of actions is used as a penalty term to prevent large actions in robot joints. We vary the action-penalty weight while keeping other weights fixed. The results are illustrated in Fig. 2. To compare across settings, the episode return excludes the action penalty term. PPO with tuned hyperparameters is highly sensitive to the penalty coefficient (e.g., in Dribbling task, the average return at the coefficient of 0.01 is roughly nine times that at 0.04). In contrast, PPO-GAGE outperforms PPO across nearly all settings. It remains robust for action penalty coefficient from 0.005 to 0.04 in every tasks. Both methods struggle when the penalty weight is too large (i.e., 0.08 in Dribbling and Balance Beam), suggesting room for future improvement. The full training curves are included in Fig. 13 and Fig. 14.

Figure 3: Sensitivity to the proxy goal estimation: we use $\hat{G} \in \{0.1, 0.25, 0.5, 1, 1.5, 2, 5, 10\}\times$ the return of the best tuned PPO (black dashed line) as the proxy goal for PPO-GAGE. We set $\sigma_0 = 0.75$ for Balance Beam and Tightrope, and 0.5 for the other three tasks.

## Unknown Individual Reward & Optimal Goal

When the agent only receives total rewards and lacks access to individual components, we approximate $g(\pi)$ via the Monte Carlo return as Eq. 1. Since the optimal return $J_{\pi^*}$ is typically unknown, we estimate it by a proxy $\hat{G}$ obtained from the best performance of a reference agent. On our IsaacLab tasks, we consider $\hat{G} \in \{0.1, 0.25, 0.5, 1, 1.5, 2, 5, 10\}\times$ the best tuned PPO return and train different PPO-GAGE agents. As depicted in Fig. 3, in four tasks, PPO-GAGE fails to learn meaningful policies when given significantly low values of $\hat{G}$, i.e. $0.1\times, 0.25\times$. Agents with other goal estimations perform at least as well as tuned PPO, and improves further with a wide range of $\hat{G}$ values for tasks such as Humanoid Cartwheel, Tightrope, and Dog Balance Beam. When using an overly large target (e.g., $\hat{G} = 10\times$), performance degrades as expected: the goal-achievement ratio remains low, so GAGE maintains a high exploration lower bound, preventing the policy from converging. This highlights a practical limitation of goal-target specification: if $\hat{G}$ is set far beyond what is attainable under the given reward and training budget, exploration may remain excessive. As a rule of thumb, we recommend choosing $\hat{G}$ within the same order of magnitude as a reasonable reference return (e.g., the best tuned baseline), and increasing it gradually (e.g., $1\times$–$2\times$) only when training reliably reaches and plateaus near the reference performance.

## Off-Policy RL with GAGE

We evaluate SAC-GAGE against SAC-v2 and MaxInfoSAC (Sukhija et al., 2025), the state-of-the-art off-policy algorithms on continuous control. Starting from the SAC-v2 implementation provided by Sukhija et al. (2025), we replace entropy regularization with the GAGE constraint and keep other components unchanged; see Algorithm 1 for details. Because tanh-squashed Gaussians can cause value overestimation and gradient vanishing near action-space boundaries, we add a policy regularizer analogous to Eq. 6 to avoid prohibitively large actor-mean magnitudes. Without access to individual reward components, we utilize an estimated total-return target $\hat{G}$ from the task-specific target return defined in (Sferrazza et al., 2024) to compute goal achievement. The hyperparameters are given in Table 5.

We conduct HPO for SAC-v2 and MaxInfoSAC over learning rates and gradient clipping values to obtain strong baselines, as detailed in Appendix C. The tuned SAC-v2 and MaxInfoSAC achieve similar performance. We evaluate on the HumanoidBench suite with 12 tasks shown in Fig. 4. Across these tasks, SAC-GAGE achieves comparable final returns on most tasks, while outperforming both baselines on H1-Hurdle-v0, H1-Highbar-Simple-v0, and H1-Stair-v0. On H1-Walk and H1-Run, SAC-GAGE additionally improves learning speed over the baselines, reaching the same final performance with only 40% of the training steps compared to tuned MaxInfoSAC, which we attribute to its initially higher exploration level.

Without the adaptive constraint on policy standard deviation applied by GAGE, SAC-v2 and MaxInfoSAC rapidly reduce the policy standard deviation $\sigma$ early in training and then keep a roughly fixed exploration level afterwards. In contrast, SAC-GAGE maintains high variance until returns improve, and then adaptively decreases it in response to goal achievement.

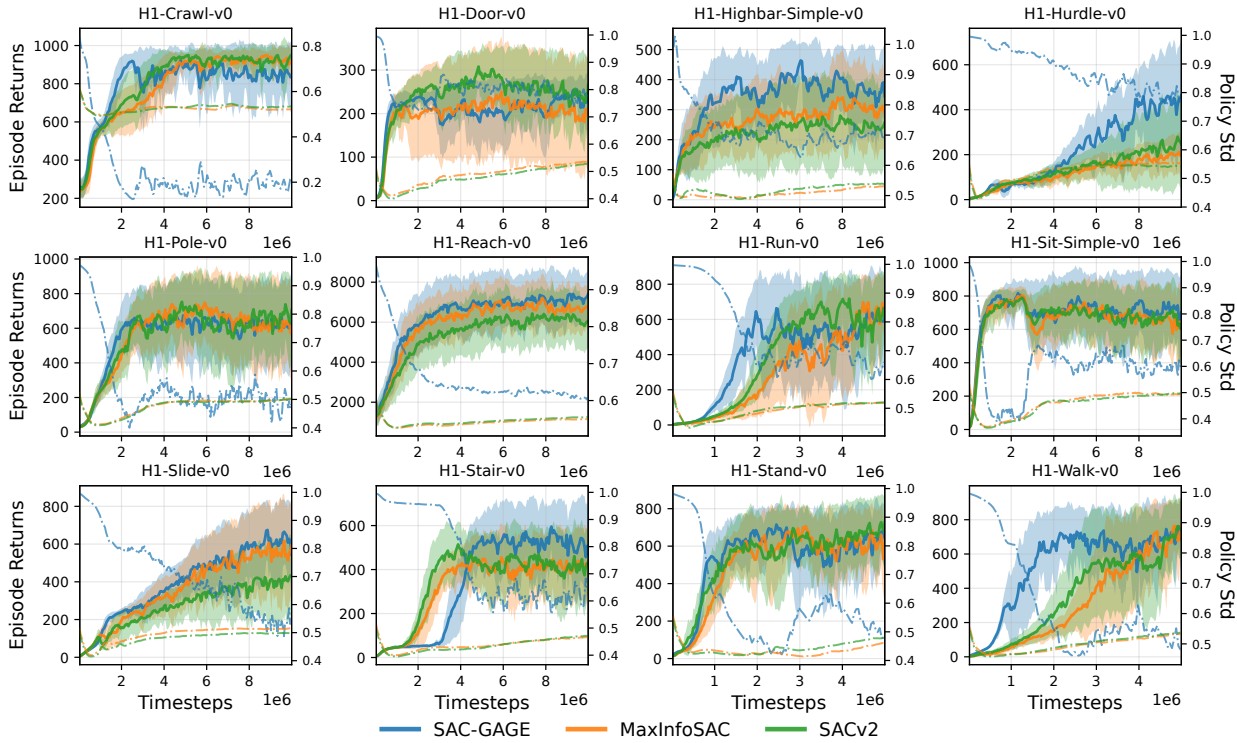

Figure 4: Comparison of tuned SAC-v2, MaxInfoSAC and SAC-GAGE on HumanoidBench. Mean episode returns (solid lines) and action $\sigma$ (dashed lines) over 5 seeds; shaded regions denote one standard deviation.

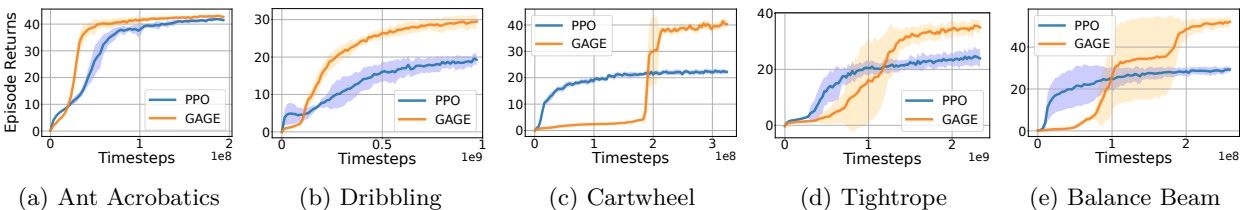

(a) Ant Acrobatics    (b) Dribbling    (c) Cartwheel    (d) Tightrope    (e) Balance Beam

Figure 5: PPO vs. PPO-GAGE on discrete action spaces with hyperpameter tuning for both. Mean episode returns and action $\sigma$ over 5 seeds; shaded regions denote one standard deviation.

## 4.2 GAGE for Discrete Action Space

We further validate GAGE with discrete policies on our IsaacLab tasks by discretizing each action dimension into 11 evenly spaced atomic actions, following Tang & Agrawal (2020). Each dimension is modeled by its own categorical distribution, yielding a factorized policy. We compare PPO-GAGE against the strongest baseline method, tuned PPO, with discrete policies. For PPO-GAGE, we apply action smoothing (see Algorithm 3) in each action dimension. The PPO entropy coefficient is tuned for each individual task. PPO-GAGE uses the other default hyperparameters from vanilla PPO with its own hyperparameters tuned. The tuned hyperparameters can be found in Table 2 and 3.

**Discrete PPO with GAGE** Fig. 5 presents the learning curves: PPO-GAGE solves all tasks, whereas tuned PPO either has worse performce or even failed in learning moving behaviors on Humanoid Cartwheel and Dog Balance Beam. Interestingly, although tuned PPO attains relatively high returns on the Humanoid Tightrope task, its goal achievement value remains low. Fig. 6 further compares the episode returns and goal achievement $g$ for PPO with the two best entropy temperatures $\beta = 5.18 \times 10^{-3}$ and $1.39 \times 10^{-2}$ alongside

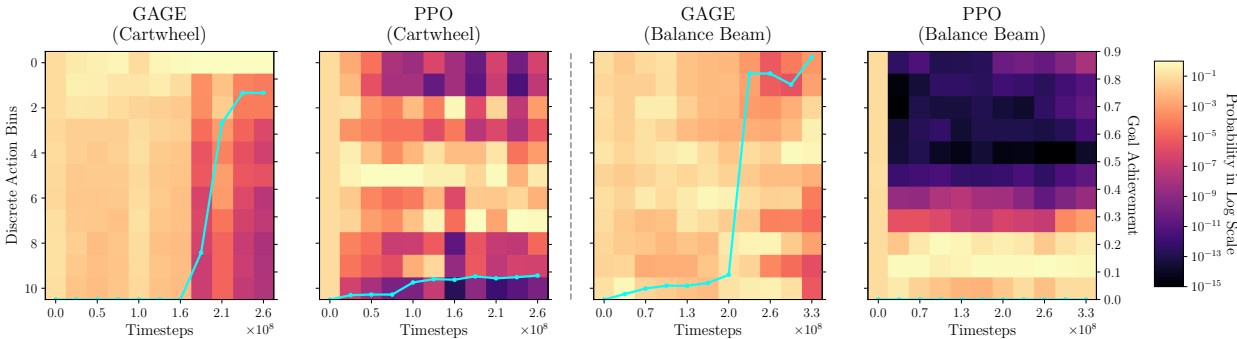

Figure 7: Comparison of policies between PPO-GAGE and PPO across two representative joints: left_thigh_x (Humanoid Cartwheel, left) and front_right_calf (Dog Balance Beam, right). The heatmaps illustrate the discrete probability distributions over 11 action bins throughout the learning process. The overlaid cyan curves indicate the Goal Achievement metric. Logarithmic scaling is applied to the probabilities to highlight the vanishingly small values in collapsed bins.

PPO-GAGE for the Humanoid Tightrope task. For $\beta = 5.18 \times 10^{-3}$, three out of five runs with differernt seeds learn to survive by remaining stationary and minimizing energy, yielding deceptively high returns. With $\beta = 1.39 \times 10^{-2}$, all agents learn to move but still converge to subopti-mal behaviors, such as unstable motion and excessive energy cost, resulting in lower returns. These observations highlight the vulnerability of entropy-based exploration to local optima, a failure mode that GAGE effectively mitigates. On Humanoid Tightrope, the final performance of GAGE with discrete policies surpasses that with continuous ones (Fig. 1), primarily due to differences in action penalties. In continuous action spaces, the unbounded nature of the PPO policy allows the agent to exploit large action magnitudes beyond the physical limits of the robot's joints. Although such actions are clipped within the simulation environment, they still incur substantial penal-ties, reducing total reward despite achieving comparable goal achievement.

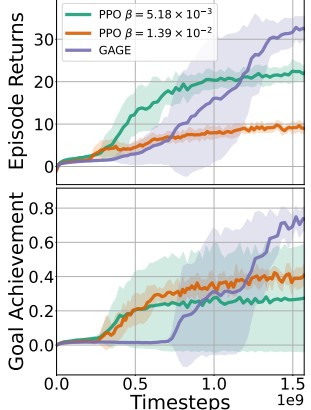

**Non-vanishing exploration** To visualize the evolution of action proba-bility distributions, we selected representative joints: *left_thigh_x* for the humanoid in Cartwheel and *front_right_calf* for the quadruped in Balance Beam. For each task, we evaluated 11 policy checkpoints saved at equal intervals throughout the learning process. By feeding identical observations to these policies and recording the probabilities for the 11 action bins of the chosen joint, we generated a $11 \times 11$ probability matrix. These matrices,

Figure 6: PPO-GAGE vs. best two PPO agents on Tightrope.

alongside the corresponding goal achievement values for GAGE and PPO agents, are displayed in Fig. 7. As illustrated, GAGE agents maintain relatively high probabilities across all action bins before securing signifi-cant goal rewards, only concentrating the distributions once performance begins to improve. This sustained variance allows the agents to explore a diverse range of behaviors to identify optimal actions. In contrast, despite failing to reach high goal achievement, PPO agents suffer from premature distribution collapse at the very beginning of the learning process. Consequently, they lose the exploratory capacity to discover optimal behaviours and become trapped in local optima. While entropy regularization prevents PPO's probabilities from collapsing into a single bin, instead spreading them across a few neighboring actions, it is insufficient to stop the remaining action bins from falling to negligibly small values.

**Continuous vs. discrete policies** Under the GAGE framework, discrete policies do not always surpass continuous ones and can even be less efficient such as on Humanoid Dribbling, Cartwheel, and Tightrope, contrary to the results of Tang & Agrawal (2020). While action discretization with entropy regularization can better mitigate local optima and premature convergence, it sacrifices the expressiveness of continuous actions and potentially optimal behaviors. The adaptive exploration strategy in GAGE allows the agent to fully leverage the continuous action space, resulting in more efficient learning.

# 5 Discussion

The results presented in this work demonstrate that Goal Achievement Guided Exploitation (GAGE) provides a principled and effective exploration scheduling strategy for addressing premature convergence in dense-reward reinforcement learning. Across diverse settings, including both discrete and continuous action spaces, on-policy and off-policy algorithms, and challenging robotic control tasks, GAGE consistently enhances learning performance.

By linking the degree of exploration directly to the agent's goal achievement, GAGE decouples exploration from manually predefined training schedules or fixed entropy targets. This allows the agent to adaptively reduce stochasticity as it approaches optimal performance. Empirically, this results in smoother convergence curves and reduced variance across random seeds compared to baselines. Notably, in highly non-convex control tasks such as Dog Balance Beam, GAGE avoids early policy collapse and continues to improve after standard methods saturate. A key strength of GAGE lies in its hard constraint formulation. Unlike entropy regularization, which indirectly promotes exploration through a soft additive term in the optimization objective, GAGE constrains the policy distribution itself through interpretable measures, the policies' standard deviation in continuous spaces and logit range in discrete spaces. This distinction leads to an advantage, ensuring that all actions maintain a lower-bounded probability throughout training, thereby preventing the irreversible loss of potentially optimal behaviors.

The experimental results highlight that GAGE is robust to hyperparameter variations and insensitive to reward shaping, two major weaknesses of standard RL algorithms. With the same set of hyperparameters, GAGE achieves superior results compared to finely tuned baselines across different tasks. This property simplifies deployment in real-world systems. Furthermore, the framework performs well even when the optimal goal value is approximated, suggesting that GAGE can be readily applied in environments with incomplete reward decomposition.

**Limitations and future work** In this work, GAGE is mainly designed for dense-reward and deterministic environments like robot control. As it guides the policy stochasticity, it belongs to undirected exploration (Thrun, 1992). In sparse-reward tasks with long horizons, its effectiveness could be limited. Despite high policy stochasticity, the agent can still struggle in getting any rewards using only random exploration. For partially observable environments or competitive multi-agent environments, deterministic policies either cannot break the tie of ambiguity or might be exploited by other agents. The implementation of GAGE in environments demanding stochastic policies is an interesting future topic. Even for the kind of environments we evaluate, GAGE offers aspects for improvement. GAGE defines the same bound for all action dimensions. It can be interesting to investigate the effect of different bounds for different dimensions. Future research can focus on improving the scalability of GAGE and applying it to more complex, dynamic, and multi-objective environments. Investigating non-linear relationships between the goal achievement and the exploration metrics could further enhance the method's adaptability to diverse RL problems.

## Acknowledgments

We thank Hao Zhu for constructive discussions during the development of GAGE. This work was partially funded by the Deutsche Forschungsgemeinschaft (DFG, German Research Foundation), Project-ID 499552394–SFB1597, and by grant numbers 428605208 and 539134284 through EFRE (FEIH_2698644) and the state of Baden-Württemberg.

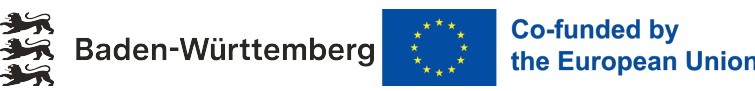

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

# A More Background and Related Work

## A.1 Premature Convergence

Premature convergence is ubiquitous in reinforcement learning due to the non-convexity of system dynamics, reward shaping, multi-objective, and function approximation error using neural networks. A policy may converge to suboptimal solutions before reaching the desired performance. We briefly clarify the differences between premature convergence and two other related problems in RL, i.e., reward hacking and the sparse reward problem.

In reward hacking, the objective function that the designer writes down admits of some "clever" solutions that formally maximize it but pervert the spirit of the designer's intent (Amodei et al., 2016). For example, a cleaning robot can close its sensors to maximize the reward for observing fewer messes. This differs from premature convergence, as agents stuck at suboptimal solutions typically do not maximize the objective function.

Sparse rewards have been regarded as one of the most challenging problems in reinforcement learning (Ladosz et al., 2022). With limited rewards scattered in the vast search space, the result is a primarily flat objective landscape. In such settings, the agent can hardly find the rewards and maximize the objective, no matter whether the policy is converged or not. This differs from premature convergence, as agents increase the objective by converging to suboptima. Of course, the challenge is exacerbated by the combination of these two issues, such as the noisy-TV problem (Burda et al., 2019). However, in this work, we focus on tasks with dense settings and numerous local optima. Note that sparse reward tasks, such as navigation and long-term robot control, can be transferred to dense reward tasks based on domain knowledge. However, auxiliary rewards often introduce new local optima and exacerbate the problem of premature convergence.

## A.2 Label Smoothing

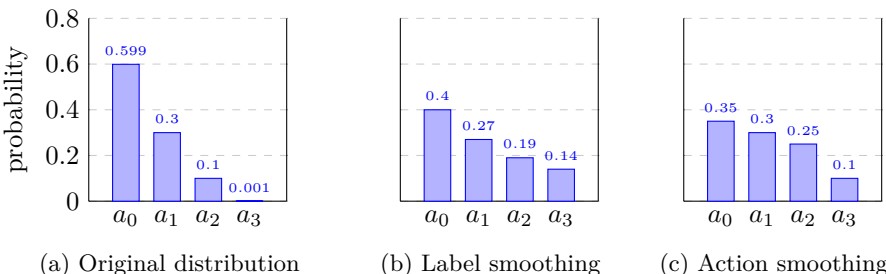

(a) Original distribution  (b) Label smoothing  (c) Action smoothing

Figure 8: A categorical distribution and its flattened results using different techniques. The resulting distributions have the same entropy 1.31. (a) Original distribution. (b) Label smoothing has a deterministic result calculated with smoothing parameter $\epsilon = 0.58$. (c) Action smoothing also has a deterministic result using temperature $\tau = 5.34$.

Label smoothing (Szegedy et al., 2016) is a popular technique for flattening discrete distributions. It is widely used in classification problems to reduce overconfidence through soft learning targets (Müller et al., 2019). It typically mixes the original distribution with a uniform distribution: $p'(a_k \mid s) = (1 - \epsilon)p(a_k \mid s) + \frac{\epsilon}{K}$, where the smoothing parameter $0 \le \epsilon \le 1$. However, it introduces a prior of uniform distribution that is inappropriate for policy learning. As shown in Fig. 8, actions with the lowest learned probabilities often lead to obvious penalties or termination states, which the agent should avoid. These actions experience the most significant increase through label smoothing, which can potentially lead to failed trials. In contrast, with action smoothing, the most significant probability increases occur for actions with middle-valued probabilities.

# B  Algorithm Implementation

We provide the pseudo code for SAC/PPO+GAGE with a Gaussian policy in Alg. [1,2] and action smoothing of a categorical policy in Alg. 3. We strike through the entropy related components in the original algorithms to highlight the differences.

---

**Algorithm 1** Soft Actor-Critic (SAC-v2) with Gaussian Policy + ~~entropy regularization~~ GAGE

---

**Require:** environment $\mathcal{E}$, discount $\gamma$, target smoothing $\tau$, replay size $N$, batch size $B$, ~~target entropy $\bar{\mathcal{H}}$ (e.g., $-\dim(\mathcal{A})$)~~, std lower bound $\sigma_{\mathrm{LB}}$, mean action factor $\alpha_{\mathrm{mean}}$, goal achievement update factor $\lambda$

1: Initialize actor $\pi_\phi(a|s) = \mathcal{N}\big(\mu_\phi(s), \Sigma_\phi(s)\big)$ (Tanh-squashed Gaussian), critics $Q_{\theta_1}, Q_{\theta_2}$, target critics $\bar{Q}_{\theta'_1} \leftarrow Q_{\theta_1}, \bar{Q}_{\theta'_2} \leftarrow Q_{\theta_2}$, ~~temperature $\alpha > 0$~~, goal achievement $g = 0$

2: Initialize replay buffer $\mathcal{D} \leftarrow \emptyset$

3: **for** episode $= 1, 2, \dots$ **do**

4:     Reset env, get $s_0$

5:     **for** t = 0,1,2,… **do**

6:         $\sigma = \overline{\sigma_\phi(s_t)} \max(\sigma_{\mathrm{LB}}, \sigma_\phi(s_t))$

7:         Sample action via reparameterization: $\epsilon \sim \mathcal{N}(0, I);\quad u = \mu_\phi(s_t) + \sigma \odot \epsilon;\quad a_t = \tanh(u)$

8:         Execute $a_t$ in $\mathcal{E}$, observe $(r_t, s_{t+1}, d_t);\quad$ store $(s_t, a_t, r_t, s_{t+1}, d_t)$ in $\mathcal{D}$

9:         **for** gradient step $= 1, 2, \dots$ **do**

10:             Sample minibatch $\{(s_i, a_i, r_i, s'_i, d_i)\}_{i=1}^B \sim \mathcal{D}$

11:             $\sigma = \overline{\sigma_\phi(s'_i)} \max(\sigma_{\mathrm{LB}}, \sigma_\phi(s'_i))$

12:             **Target action and log-prob:** $\epsilon' \sim \mathcal{N}(0, I); u' = \mu_\phi(s'_i) + \sigma \odot \epsilon'; a'_i = \tanh(u');$ compute $\log \pi_\phi(a'_i|s'_i)$ with tanh correction

13:             **Target Q:** $\hat{Q}_i = \min\big(\bar{Q}_{\theta'_1}(s'_i, a'_i), \bar{Q}_{\theta'_2}(s'_i, a'_i)\big)$

14:             **Bellman target:** $y_i = r_i + \gamma(1 - d_i)\big(\hat{Q}_i - \overline{\alpha \log \pi_\phi(a'_i|s'_i)}\big)$

15:             **Critic loss:** $\mathcal{L}_Q(\theta_j) = \frac{1}{|B|} \sum_i \big(Q_{\theta_j}(s_i, a_i) - y_i\big)^2, \quad j \in \{1, 2\}$

16:             $\sigma = \overline{\sigma_\phi(s_i)} \max(\sigma_{\mathrm{LB}}, \sigma_\phi(s_i))$

17:             Update $\theta_1, \theta_2$ by minimizing $\mathcal{L}_Q$

18:             **Actor sampling:** $\epsilon \sim \mathcal{N}(0, I); u = \mu_\phi(s_i) + \sigma \odot \epsilon; a_i = \tanh(u);$ compute $\log \pi_\phi(a_i|s_i)$

19:             **Actor loss (reparameterized) augmented by mean action regularization:**

$$\mathcal{L}_\pi(\phi) = \frac{1}{|B|} \sum_i \left(\overline{\alpha \log \pi_\phi(a_i|s_i)} - \min_{k=1,2} Q_{\theta_k}(s_i, a_\phi(s_i)) + \alpha_{\mathrm{mean}} \sum_i \|\mu_\phi(s_i)\|_2^2\right)$$

20:             Update $\phi$ by minimizing $\mathcal{L}_\pi$

21:             ~~**Temperature loss (optional tuning):**~~

$$\overline{\mathcal{L}_\eta(\eta) = \frac{1}{|B|} \sum_i \eta\big(-\log \pi_\phi(a_i|s_i) - \bar{\mathcal{H}}\big)}$$

22:             ~~Update $\alpha$ by minimizing $\mathcal{L}_{\overline{\alpha}}$~~

23:             **Target update:** $\theta'_j \leftarrow \tau \theta_j + (1 - \tau)\theta'_j, \quad j \in \{1, 2\}$

24:         **end for**

25:         **if** $d_t$ **then**

26:             Update goal achievement: $g \leftarrow \lambda g + (1 - \lambda)g_\mathrm{e}$, where $g_\mathrm{e}$ is the episode goal achievement

27:             Update the $\sigma_{\mathrm{LB}}$ based on the agent's performance: $\sigma_{\mathrm{LB}} \leftarrow \sigma_{\mathrm{LB}}(1 - g)$

28:             **break**

29:         **end if**

30:         $s_t \leftarrow s_{t+1}$

31:     **end for**

32: **end for**

---

---

**Algorithm 2** Proximal Policy Optimization (PPO) Algorithm with Gaussian Policy + ~~entropy~~ GAGE

---

1: Initialize policy mean parameters $\theta_0$, policy standard deviation $\sigma_0$, value function parameters $\phi_0$, goal achievement $g_0 = 0$, goal achievement update factor $\lambda$
2: **for** iteration $k = 0, 1, 2, \ldots$ **do**
3:    Collect set of trajectories $\{(s_t, a_t, r_t, s_{t+1})\}$ by running policy $\pi_{\theta_k}(a_t|s_t) = \mathcal{N}(\mu_{\theta_k}(s_t), \sigma_k^2)$ in the environment
4:    **for** each time step $t$ **do**
5:      Compute advantage estimates $\hat{A}_t$ based on value function $V_{\phi_k}(s_t)$
6:    **end for**
7:    Update the policy by maximizing the PPO-CLIP objective with an added entropy term:

$$\theta_{k+1}, \sigma_{k+1} = \arg\max_{\theta,\sigma} \mathbb{E}_t \Big[ \min \left( \frac{\mathcal{N}(\mu_\theta(s_t), \sigma^2)}{\mathcal{N}(\mu_{\theta_k}(s_t), \sigma_k^2)} \hat{A}_t, \text{clip} \left( \frac{\mathcal{N}(\mu_\theta(s_t), \sigma^2)}{\mathcal{N}(\mu_{\theta_k}(s_t), \sigma_k^2)}, 1 - \epsilon, 1 + \epsilon \right) \hat{A}_t \right)$$

$$+ \beta H(\pi_\theta(a_t|s_t)) \Big]$$

where $\mu_{\theta_k}(s_t)$ is the mean of the Gaussian action distribution, $\sigma_k$ is the standard deviation (separately learned), and $H(\pi_\theta(a_t|s_t))$ is the entropy of the policy, encouraging exploration. The term $\beta$ controls the weight of the entropy regularization.
8:    Update the value function by minimizing the following loss:

$$\phi_{k+1} = \arg\min_\phi \mathbb{E}_t \left[ (V_\phi(s_t) - R_t)^2 \right]$$

9:    Update the goal achievement: $g_{k+1} = \lambda g_k + (1 - \lambda)R$
10:    Lower bound the standard deviation parameter $\sigma$ based on the agent's performance:

$$\sigma_{k+1} = \max(\sigma_{k+1}, -\sigma_0 g_k + \sigma_0)$$

11: **end for**

---

**Algorithm 3** Action Smoothing Algorithm

---

**Require:** Network outputs $\{z_1, z_2, \ldots, z_K\}$, goal achievement $g(\pi)$, hyperparameters $\delta_{z,0}$, $\delta_{z,1}$
**Ensure:** Action probabilities $p(a_k \mid s)$
1: **Calculate the original logit difference**:

$$\delta_z = \max_k(z_k) - \min_k(z_k).$$

2: **Compute the upper-bound** for the logit difference:

$$\delta_{z,\text{UB}} = (\delta_{z,1} - \delta_{z,0})g(\pi) + \delta_{z,0}.$$

3: **Calculate the temperature**:

$$\tau(g, z) = \frac{\max(1, \delta_z)}{\delta_{z,\text{UB}}}.$$

4: **Compute action probabilities** using softmax with temperature:

$$p(a_k \mid s) = \frac{\exp(z_k/\tau(g,z))}{\sum_{i=1}^K \exp(z_i/\tau(g,z))}, \quad \text{for } k = 1, 2, \ldots, K.$$

---

## C  Additional Results

In this section, we provide more detailed experiment results with continuous action spaces.

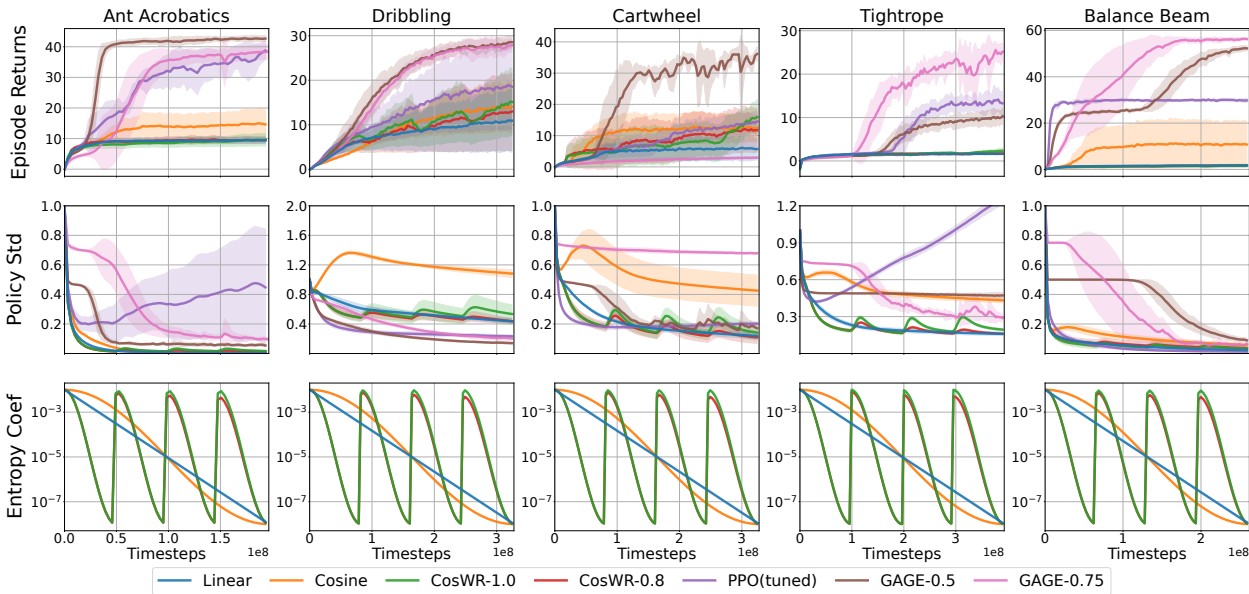

Figure 9: Additional training results of experiments with continuous action space with differerent entropy coefficient schedules. We plot the mean over 5 seeds. The faint area represents one standard deviation

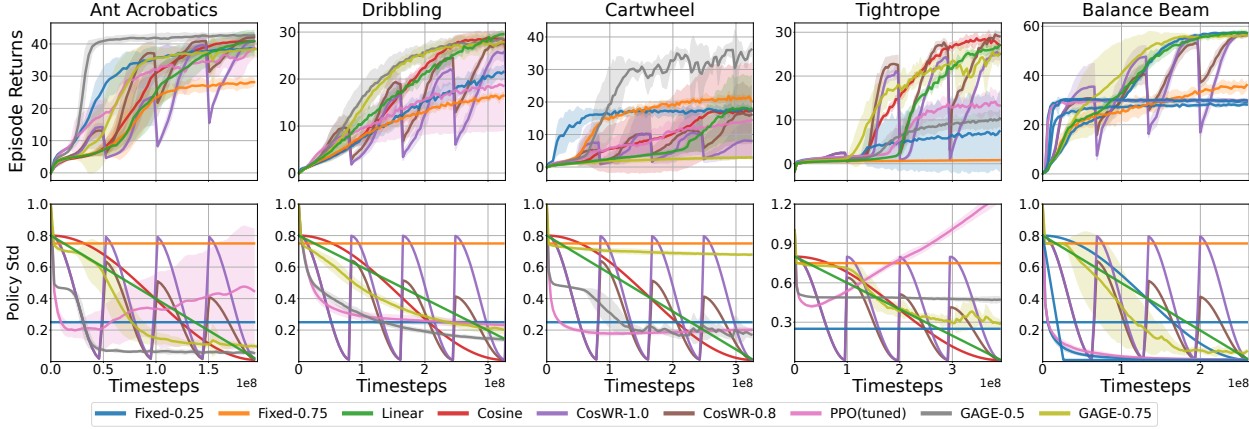

Figure 10: Additional training results of experiments with continuous action space with differerent $\sigma$-schedules. We plot the mean over 5 seeds. The faint area represents one standard deviation

**Std Scheduling vs. Entropy Coefficient Scheduling**   In continuous-action PPO, we implement GAGE by directly regulating the Gaussian policy standard deviation $\sigma$, rather than by applying a schedule to the entropy coefficient $\alpha$. To justify this choice, we perform a controlled ablation that applies the same family of common hand-designed time schedules (linear decay, cosine decay, and cosine annealing with warm restarts) to either entropy coefficient or policy std $\sigma$, while keeping all other PPO settings fixed. We consider Linear and Cosine decay, which decrease a parameter from a specified start value to an end value over the full training budget, as well as cosine annealing with warm restarts, denoted CosWR-1.0 and CosWR-0.8. The warm-restart variants partition training into four equal cycles; in each cycle, the value is reset to the start and annealed to the end, with an optional restart-amplitude decay of 0.8 across cycles

(no decay for CosWR-1.0). Concretely, all entropy-coefficient schedules start at $\alpha = 0.01$ and decay in log scale to $\alpha = 10^{-8}$ (effectively zero), while all std schedules start at $\sigma = 0.8$ and decay in linear scale to $\sigma = 0$; for clarity, our PPO (tuned) baseline uses a fixed entropy coefficient, selected as the best-performing constant $\alpha$. The results reveal a clear asymmetry: in our setting, entropy-coefficient scheduling does not help, every manually designed entropy coefficient schedule underperforms the best fixed (tuned) entropy coefficient. In contrast, std scheduling provides a meaningful and reliable exploration-control interface: several hand-designed $\sigma$ schedules are competitive, and cosine decay in particular often matches GAGE on individual tasks. Nevertheless, GAGE achieves the best overall performance across the suite, providing a single adaptive schedule that is consistently strong without task-specific schedule selection. Moreover, the std trajectories discovered by GAGE are qualitatively similar to the top-performing manual std schedules; for example, on the Cartwheel task, GAGE learns a decay profile that lies between the tuned fixed-$\alpha$ baseline and the linear/cosine std baselines, yielding better performance. Overall, these findings support our design decision to control exploration at the distribution level via $\sigma$, rather than indirectly through objective-level weighting via the entropy coefficient.

**Details of Population-based Training**   To provide a performance-based, automated alternative to hand-designed entropy schedules, we additionally include a Population-Based Training (PBT) (Dembrower et al., 2020) baseline that tunes the PPO entropy coefficient online using population performance. In each PBT run, we maintain a population of 8 policies trained concurrently and periodically apply selection and mutation, using each member's episode return as the optimization objective. To match the overall training budget of our standard runs, the population shares a fixed total amount of environment interaction: we run 1024 environments in parallel and allocate 1024/8 environments to each policy, so that the population collectively consumes the same number of environment steps as a single-policy run. PBT updates are performed at a fixed interval equal to 1/100 of the total training horizon (the same fraction for all tasks). At each update, underperforming members are replaced by better-performing ones (exploit), and their entropy coefficients are perturbed with probability 0.25 (explore) by multiplying them with a random factor sampled uniformly from $[1.1, 2.0]$. Mutations are applied only when the population has sufficiently diverged in performance, controlled by dispersion thresholds with $\theta_{std} = 0.1$ and $\theta_{abs} = 0.025$. The initial entropy coefficients are sampled in log space from $10^{-6}$ to $10^{-3}$. Reported PBT results are averaged over 5 independent PBT runs with different random seeds. We follow the implementation of Petrenko et al. (2023) and refer readers there for additional details.

**Intrinsic Reward Weight**   To evaluate the effect of intrinsic rewards in the proposed challenging control tasks, we trained several RND agents using different weight combinations for extrinsic and intrinsic rewards: (2.0, 1.0), (2.0, 0.5), (1.0, 1.0), (1.0, 2.0), and (1.0, 4.0). The weight values (2.0, 1.0) are consistent with those used in the original RND work (Burda et al., 2019) and subsequent research (Yang et al., 2024). Therefore, we also used this ratio for the experiments presented in Fig. 1. As shown in Fig. 11, none of the RND agents succeeded in solving the task. Agents with larger ratios of extrinsic-to-intrinsic weights exhibited learning patterns similar to standard PPO, which does not use intrinsic rewards. As the ratio decreased, the agents focused more on exploring novel states, as indicated by larger standard deviations during training. However, this increased exploration did not contribute to solving the task. Instead, the novelty-based exploration resulted in decreased extrinsic rewards. This phenomenon highlights the distinct focus of our work compared to novelty-based exploration methods. Our work focuses on addressing premature convergence, an issue that is equally important but has been largely overlooked until now. In contrast, curiosity-based methods primarily tackle sparse rewards. The difference in focus is also reflected in the existing benchmarks for exploration algorithms. Most environments are designed with sparse rewards and moderate local optima, which can be effectively addressed using novelty-based exploration. For example, environments like Fetch (Plappert et al., 2018), MiniGrid (Chevalier-Boisvert et al., 2023), AntMaze, and Adroit manipulation tasks (Fu et al., 2020) are "safe," with sparse termination states or penalties distributed across the state space. Agents can easily avoid termination and penalty states while exploring for rewards. In such environments, exploring unseen states is a highly effective strategy. However, novelty-based methods struggle in scenarios with more severe local optima. For instance, Noisy-TV has been recognized as a major issue for novelty-based methods, even though it only involves local optima introduced by environment stochasticity. The challenges posed by more severe local optima have not yet been fully explored. In this

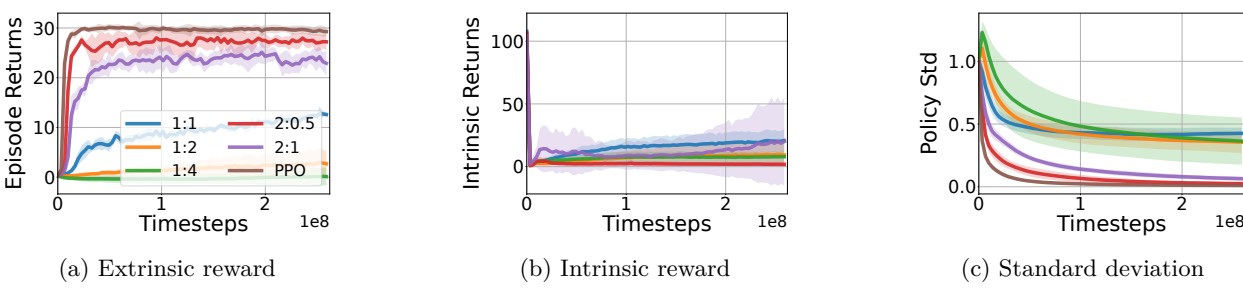

(a) Extrinsic reward        (b) Intrinsic reward        (c) Standard deviation

Figure 11: Investigating the effect of novelty-based intrinsic reward to the learning of Dog Balance Beam task. The curves with legend 1:2 represent the agent trained using extrinsic and intrinsic coefficients of (1.0, 2.0).

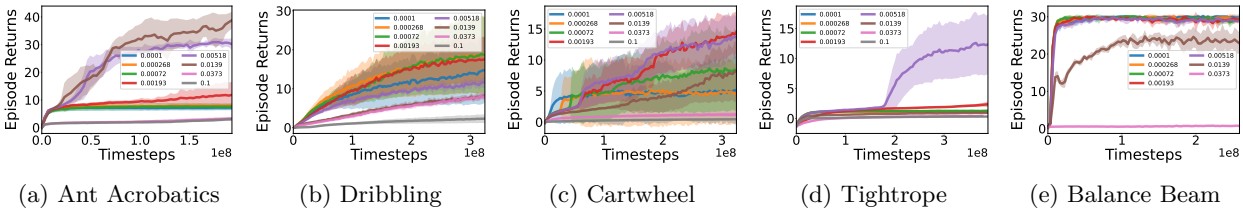

(a) Ant Acrobatics    (b) Dribbling    (c) Cartwheel    (d) Tightrope    (e) Balance Beam

Figure 12: Different Fixed Entropy Coefficient for Continuous Control Tasks using PPO

work, we aim to push the boundaries of RL exploration research into environments with more challenging local optima issues. The proposed IsaacLab tasks reflect real-world robot control scenarios where optimal behaviors occupy only a small portion of the state space, while most of the state space leads to penalties such as falling down or wasting energy. This dominant penalizing space creates challenging local optima. In such environments, novelty-based exploration often results in sampling mostly failed trajectories and becoming trapped in local optima.

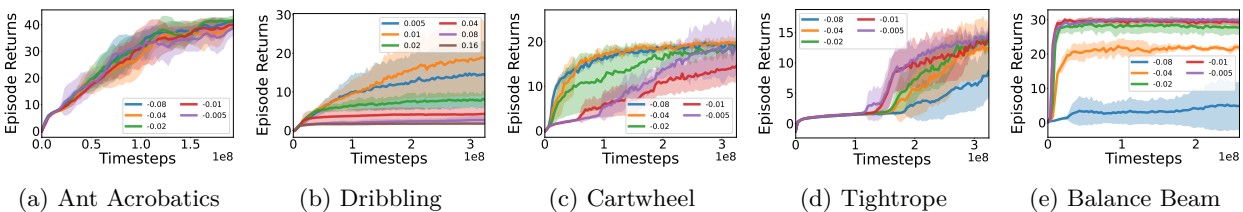

(a) Ant Acrobatics    (b) Dribbling    (c) Cartwheel    (d) Tightrope    (e) Balance Beam

Figure 13: Training curves of different L2 action penalty coefficients for PPO in continuous control tasks

# D  Experimental Details

## D.1  Tasks Setup

We design five challenging continuous control tasks in IsaacLab. Three robots with many degrees of freedom learn challenging locomotion or dynamic manipulation behaviors. The robots include a humanoid robot with 21 joints, a dog robot (Unitree Go2) with 12 joints, and an ant robot with 8 joints. In Table 1, we provide the reward composition of different tasks.

**Humanoid Tightrope (HT)**  The humanoid robot learns side walking for 2m/s on a tightrope, i.e., a cylindrical bar with a diameter of only 0.1m. This is more challenging than walking forward because balancing with two arms stretched to both sides would be more difficult.

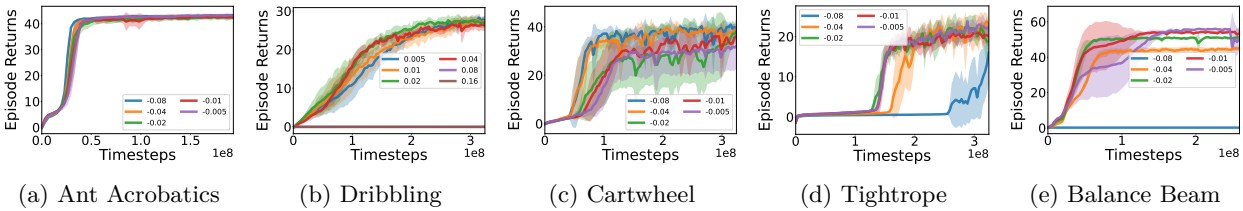

Figure 14: Training curves of different L2 action penalty coefficients for GAGE in Continuous Control Tasks

Table 1: Reward weights of continuous control tasks. The rewards and penalties from left to right are for desired locomotion velocity, environment not terminating, robot orientation, robot distance to the manipulated object, large action commands, energy consumption, joint position too close to limitations, robot velocity perpendicular to the desired direction, object velocity perpendicular to the desired direction, joint torque, joint acceleration, and action changing rate. The selected goal reward for goal achievement calculation is marked in green background.

|  | reward | | | | penalty | | | | | | | |
|---|---|---|---|---|---|---|---|---|---|---|---|---|
|  | $v_x$ | alive | orient | $d_{\text{obj}}$ | $\|a\|_2^2$ | $E$ | $\theta_{\text{limit}}$ | $v_y$ | $v_{y,\text{obj}}$ | $T$ | $\ddot{\theta}$ | $\dot{a}$ |
| HT | 0.5 | 1.0 | 1.0 | 0 | 0.01 | 0.05 | 0.25 | 1.0 | 0 | 0 | 0 | 0 |
| HD | 0.3 | 0.4 | 1.0 | 0.2 | 0.01 | 0.01 | 0.25 | 0 | 0.5 | 0 | 0 | 0 |
| HC | 2.0 | 1.0 | 0.5 | 0 | 0.01 | 0.05 | 0.25 | 0 | 0 | 0 | 0 | 0 |
| DB | 1.0 | 1.0 | 1.0 | 0 | 0.005 | 0 | 0 | 1.0 | 0 | 1e-6 | 2.5e-8 | 0.001 |
| AA | 1.0 | 1.0 | 1.0 | 0 | 0.005 | 0.05 | 0.1 | 0 | 1.0 | 0 | 0 | 0 |

**Humanoid Dribbling (HD)**   The humanoid robot learns to dribble a football at a high speed (3.5m/s). Additionally, the robot gets random commands for turning the target direction for up to $\frac{\pi}{4}$rad.

**Humanoid Cartwheel (HC)**   The humanoid robot learns to perform cartwheel at a speed of 6rad/s. It requires precisely coordinated movement of the whole body, which is more difficult than normal locomotion tasks, which primarily rely on the lower body of the robot.

**Dog Balance Beam (DB)**   The dog robot learns to walk on a balance beam at a speed of 2m/s. The beam has a square cross-section with 0.1m side length. Moreover, the balance beam is tilted for $\frac{\pi}{9}$rad so that the robot has to climb a slope while balancing.

**Ant Acrobatics (AA)**   The ant robot with four legs learns to balance a pole vertically on its torso while standing on a ball. The pole has a length of 2m. The ball has a diameter of the same value. Moreover, the robot has to learn to roll the ball forward at a target speed of 1m/s.

### D.2   Hyperparameters Optimization and Analysis

GAGE is a principled framework which can be integrated into different algorithms. Due to the differences among different algorithms and action spaces, GAGE introduces different sets of hyperparameters to ensure the lower bound of the distribution flatness. In this part, we show the details of hyerparameter optimization in each case and perform an analysis of GAGE's hyerparameters as a reference for practitioner.

**Default Hyperparameter for PPO**   We follow the implementation in RSL-RL[1] PPO and reuse their default hyperparameters in the vanilla PPO for continuous control tasks. GAGE shares hyperparameters with vanilla PPO except for its own hyperparameters. We include the default parameters in Table 4.

---

[1] https://github.com/leggedrobotics/rsl_rl

Table 2: Tuned hyperparameters for PPO on different tasks

|  | Action Type | AA | HD | HC | HT | DB |
|---|---|---|---|---|---|---|
| entropy coef | continuous | 1.39e-2 | 7.2e-4 | 1.93e-3 | 5.18e-3 | 1e-4 |
|  | discrete | 5.18e-3 | 5.18e-3 | 1.93e-3 | 5.18e-3 | 5.18e-3 |

**Hyperparameter Optimization for PPO**  GAGE is addressing the premature convergence by adaptively changing the lower bound of the policies' standard deviation. To obtain a strong baseline to compare in our tasks, we perform a grid search for the entropy coefficient of each continuous or discrete task. We evaluate 8 different entropy coefficients uniformly in log-scale from $1 \times 10^{-4}$ to 0.1 with 5 seeds. The top-performing hyperparameter setting will then be used for the tuned PPO baseline. We select the best-performing hyperparameter by averaging the last episode return over 5 runs with different seeds. Detailed training curves can be found in Fig. 12. The resulting entropy coeffcients are summarized in Table 2.

**Hyperparameter Optimization and Analysis for continuous PPO-GAGE**  We use PPO as the backbone for all IsaacLab experiments. Our implementation is based on `rsl_rl` v2.0.0, with the only algorithmic modification being the action-smoothing procedure described in Algorithm 3. For PPO-GAGE, we disable the entropy term by setting the temperature to zero; aside from this change, we keep all hyperparameters identical across methods to ensure a fair comparison (see Table 4). As shown in Fig. 1, in continuous-action settings a single default choice of $\sigma_0 = 0.5$ performs well across most tasks and is generally sufficient. The only exception is the Balance Beam task, where $\sigma_0 = 0.75$ yields slightly better asymptotic performance. Based on these results, we recommend $\sigma_0 = 0.5$ as a practical default for new downstream tasks, with $\sigma_0 = 0.75$ as a reasonable alternative when stronger late-stage exploration is beneficial.

**Hyperparameter for discrete PPO-GAGE**  We discretize each action dimension into 11 bins, while keeping all other hyperparameters identical to the continuous PPO implementation. Discrete PPO-GAGE introduces three method-specific hyperparameters, $(\delta_{z,0}, \delta_{z,1}, \alpha_{\text{logit}})$. For each environment, we perform a grid search over $\delta_{z,0} \in \{-3, -4, -5\}$, $\delta_{z,1} \in \{-10, -15, -20\}$, and $\alpha_{\text{logit}} \in \{0.001, 0.01, 0.1\}$. For clarity, the ablation in Fig. 15 fixes $\delta_{z,1} = -20$, as we found it to be less influential than the other two parameters. Overall, both $\delta_{z,0}$ and $\alpha_{\text{logit}}$ noticeably affect performance. The best-performing hyperparameter settings are reported in Table 3. On simpler tasks (e.g., Ant Acrobatics), performance is relatively insensitive across the grid. In contrast, on tasks that require stronger exploration (e.g., Balance Beam and Cartwheel), larger $\alpha_{\text{logit}}$ tends to yield better performance. As a practical guideline, we recommend starting with a smaller $\alpha_{\text{logit}}$ and increasing it if learning plateaus early or converges to suboptimal behavior. Regarding the initial exploration bound $\delta_{z,0}$, a less negative value (e.g., $-3$) generally provides sufficient stochasticity to navigate the reward landscape and discover optimal behavioral modes in stable environments. However, the optimal value is linked to the stability of the task dynamics. In highly unstable scenarios, such as the Dog Balance Beam task, practitioners should prioritize a more negative $\delta_{z,0}$ (e.g., $-5$). This more constrained initial bound prevents the agent from frequently sampling "dangerous" actions that lead to immediate failure, thereby ensuring the agent survives long enough before transitioning to high-velocity gaits.

Table 3: Tuned hyperparameters for PPO-GAGE on different tasks with discrete actions

|  | AA | HD | HC | HT-F | DB |
|---|---|---|---|---|---|
| $\delta_{z,0}$ | -4 | -4 | -3 | -4 | -5 |
| $\delta_{z,1}$ | -20 | -20 | -20 | -15 | -15 |
| $\alpha_{\text{logit}}$ | 0.1 | 0.01 | 0.1 | 0.01 | 0.01 |

**Hyperparameter for SAC-v2, SAC-GAGE, and MaxinfoSAC**  For the HumanoidBench experiments, we build on the public implementation used by Sukhija et al. (2025)[2]. To ensure a fair comparison,

---

[2]https://github.com/sukhijab/maxinforl_jax

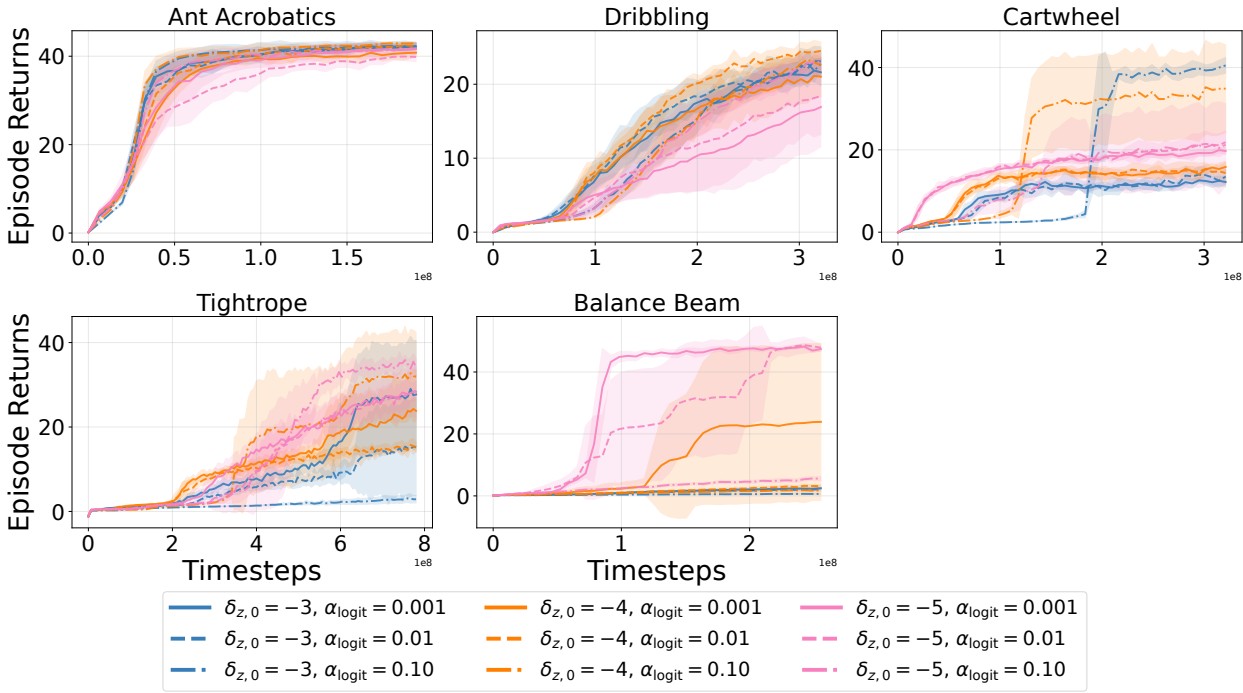

Figure 15: Hyperparameter analysis of discrete PPO-GAGE. We plot the mean over 5 seeds. The faint area represents one standard deviation. We fixed the $\delta_{z,1} = -20$ as the differences due to this value are negligible from our empirical experience.

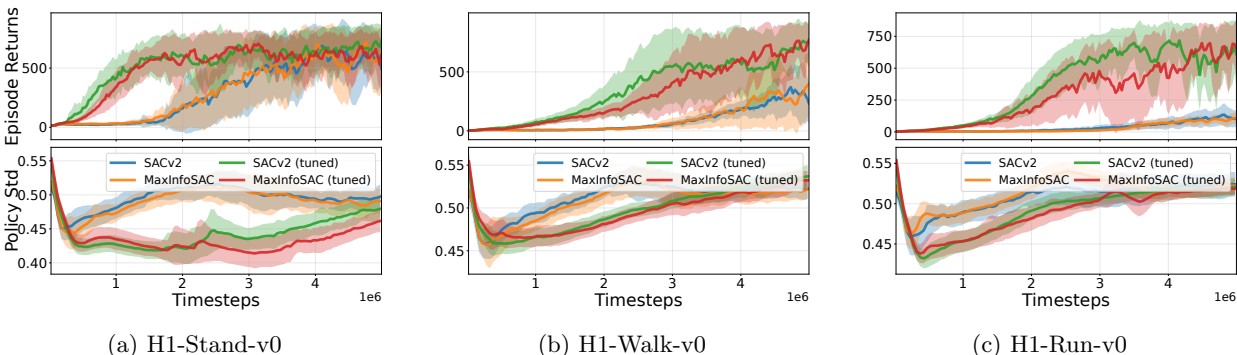

(a) H1-Stand-v0          (b) H1-Walk-v0          (c) H1-Run-v0

Figure 16: Comparison of the original SAC-v2, MaxInfoSAC and tuned versions on a selection of Humanoid-Bench. Mean episode returns and action $\sigma$ over 5 seeds; shaded regions denote one standard deviation.

we follow the hyperparameters reported by the original authors for the same tasks (H1-Stand-v0, H1-Walk-v0, and H1-Run-v0; see Table 5). For SAC-GAGE, we disable the entropy term by setting the temperature to zero. In addition, we tuned two generally applicable optimization hyperparameters, the critic learning rate and gradient-norm clipping, and found this substantially improved performance, reducing the critic learning rate from $5 \times 10^{-4}$ to $1 \times 10^{-4}$ and enabling gradient-norm clipping with threshold 1.0 (previously disabled). The effect of this tuning is shown in Fig. 16: tuned SAC and MaxInfoSAC improve performance by roughly $2\times$ on H1-Stand-v0 and H1-Walk-v0 and by about $5\times$ on H1-Run-v0. SAC-GAGE introduces three method-specific hyperparameters ($\sigma_0, \alpha_{\text{mean}}, \hat{G}$). For all SAC-GAGE results reported in Fig. 4, we use a single hyperparameter setting across the task set to avoid per-task tuning: $\sigma_0 = 1.0$ and $\alpha_{\text{mean}} = 0.01$, selected as the best overall setting on the same tasks considered by MaxInfoRL. For the proxy goal estimation $\hat{G}$, we utilize the task-specific target returns provided by Sferrazza et al. (2024); for simplicity, we set $\hat{G}$ to $1.5\times$ the provided target return for all HumanoidBench tasks. To analyze hyperparameter sensitivity, we additionally conduct a grid search over $\sigma_0 \in \{0.75, 1.0, 1.25\}$ and $\alpha_{\text{mean}} \in \{0.001, 0.01, 0.1\}$ across all tasks.

As shown in Fig. 17, SAC-GAGE is relatively insensitive within this range, with performance degrading mainly when $\alpha_{\text{mean}} = 0.1$. Based on these results, we use $\sigma_0 = 1.0$ and $\alpha_{\text{mean}} = 0.01$ as default values for all tasks, providing a practical starting point and indicating that SAC-GAGE can perform well out of the box.

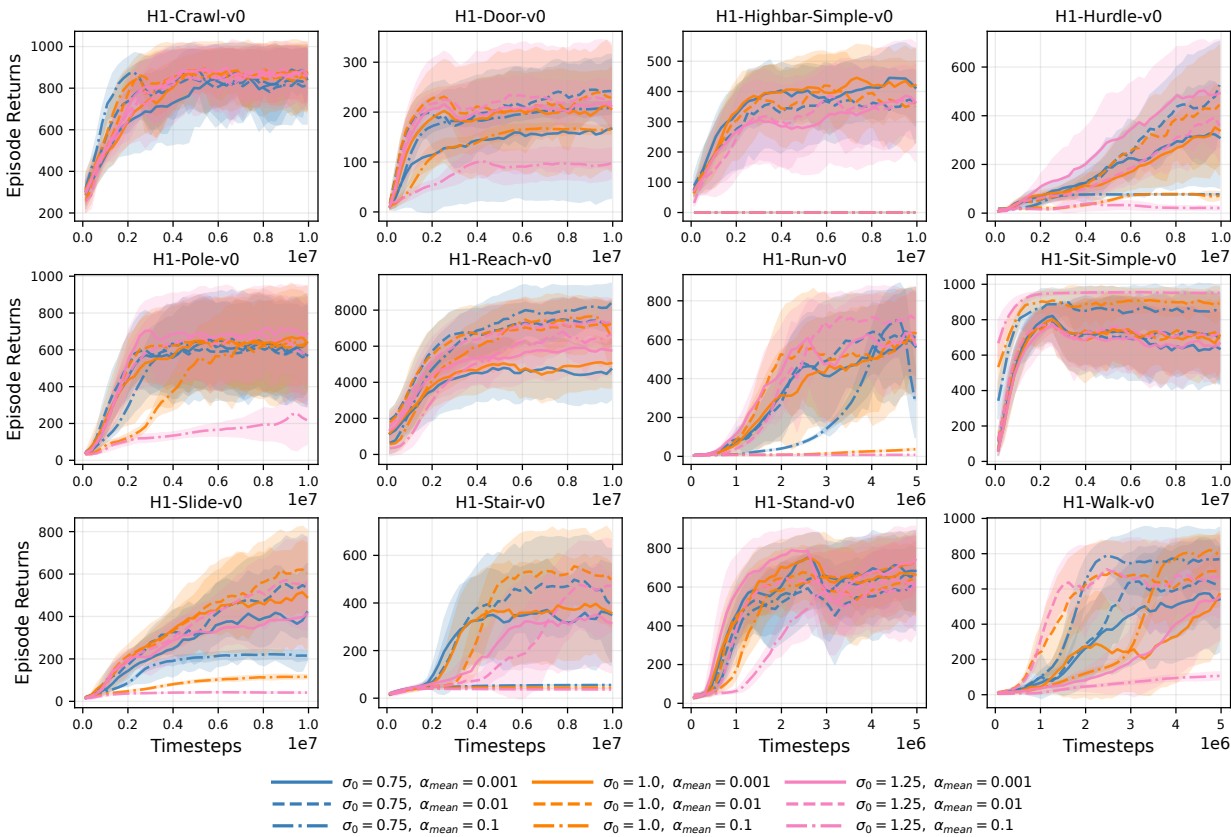

Figure 17: Hyperparameter analysis of SAC-GAGE. We plot the mean episode return over 5 seeds. The faint area represents one standard deviation.

Table 4: Common hyperparameters for PPO(-GAGE)

| Hyperparameter | Value |
|---|---|
| **Algorithm** | |
| Value loss coefficient | 1.0 |
| Clip parameter ($\epsilon$) | 0.2 |
| Use clipped value loss | True |
| Desired KL divergence | 0.01 |
| Discount factor ($\gamma$) | 0.99 |
| GAE parameter ($\lambda$) | 0.95 |
| Max gradient norm | 1.0 |
| Learning rate | 0.0005 |
| Number of learning epochs | 5 |
| Number of mini-batches | 4 |
| Learning rate schedule | Adaptive |
| **Policy** | |
| Activation function | ELU |
| Actor hidden dimensions | [400, 200, 400] |
| Critic hidden dimensions | [400, 200, 100] |
| Initial noise standard deviation | 1.0 |
| **Runner** | |
| Number of steps per environment | 24 |
| Number of environments | 1024 |
| Empirical normalization | False |
| **RND** | |
| Intrinsic Reward coefficient | 1 |
| Extrinsic Reward coefficient | 2 |
| Intrinsic Reward Normalization | yes |

Table 5: Hyperparameters for SAC & MaxInfoSAC & SAC-GAGE

| Hyperparameter | SAC/tuned | MaxInfoSAC/tuned | SAC-GAGE |
|---|---|---|---|
| hidden dimensions | [512, 512] | [512, 512] | [512, 512] |
| discount ($\gamma$) | 0.99 | 0.99 | 0.99 |
| tau | 0.005 | 0.005 | 0.005 |
| target update period | 1 | 1 | 1 |
| target entropy | $-\dim(\mathcal{A})$ | $-\dim(\mathcal{A})$ | - |
| backup entropy | true | true | - |
| actor lr | 0.0005 | 0.0005 | 0.0005 |
| critic lr | 0.0005/0.0001 | 0.0005/0.0001 | 0.0001 |
| temperature lr | 0.0005 | 0.0005 | - |
| initial temperature | 1.0 | - | - |
| ensemble lr | - | 0.0005 | |
| information gain temperature lr | - | 0.0005 | - |
| gradient norm clip | None/1.0 | None/1.0 | 1.0 |
| initial std lower bound $\sigma_0$ | - | - | 1.0 |
| mean action factor $\alpha_{\mathrm{mean}}$ | - | - | 0.01 |
| proxy goal estimation $\hat{G}$ | - | - | 1.5×benchmark value |

Table 6: HumanoidBench task-specific target return (Sferrazza et al., 2024)

| Crawl | Door | Highbar | Hurdle | Pole | Reach | Run | Sit | Slide | Stair | Stand | Walk |
|---|---|---|---|---|---|---|---|---|---|---|---|
| 700 | 600 | 750 | 700 | 700 | 12000 | 700 | 750 | 700 | 700 | 800 | 700 |

