# OpenReview forum: "Goal Achievement Guided Exploitation: A Principled Performance-Based Scheduling Framework for Reinforcement Learning"
_TMLR — Accepted by TMLR_

### Review · Reviewer_PA8K · 2025-12-18

**Summary Of Contributions:**

This submission proposes a reinforcement-learning framework called Goal Achievement Guided Exploitation (GAGE) that aims to mitigate premature policy convergence by linking how much the policy is allowed to collapse to an explicit goal-achievement measure. Goal achievement is defined as a ratio between the current policy’s expected return and an optimal (or maximum) expected return, with a practical refinement that measures goal achievement using goal-reward terms while excluding auxiliary shaping rewards, and approximates it with a moving average over per-episode ratios. The framework replaces maximum-entropy regularization with adaptive hard constraints on policy flatness: lower-bounding Gaussian policy standard deviation for continuous actions, and lower-bounding a discrete policy’s action probabilities via logit-range–based smoothing for discrete actions, with the intent of guaranteeing non-vanishing exploration until near-goal performance is reached. The paper instantiates GAGE in on-policy PPO (PPO-GAGE) and off-policy settings, and emphasizes fair comparisons by disabling the MaxEnt term (entropy temperature set to zero) while keeping other backbone hyperparameters aligned with the baseline where applicable.

**Audience:**

Yes

**Audience Explanation:**

The submission targets a central design choice in modern reinforcement learning—maximum-entropy regularization—and offers an alternative mechanism that is simple to integrate with common algorithms (e.g., PPO and SAC) and applicable to both continuous and discretized action parameterizations. Researchers working on exploration, policy collapse/premature convergence, and practical robustness in robot control (including reward-shaping sensitivity) are likely to find the perspective and results interesting.

**Broader Impact Concerns:**

None identified beyond standard considerations for deploying exploratory RL in the physical world; the paper’s experiments are in simulation.

**Claims And Evidence:**

Yes

**Claims Explanation:**

The core claims are supported by a clear motivation, formal definitions of goal achievement and the associated exploration constraint, and empirical results on both on-policy and off-policy backbones. In particular, the paper motivates why entropy regularization can both change the optimization objective and fail to guarantee sufficient exploration, then proposes a goal-achievement-conditioned hard constraint that yields interpretable control of exploration. The experiments on challenging robot-control tasks and HumanoidBench generally align with the stated claims, and the appendix provides useful implementation and tuning details. That said, reporting a few benchmark-specific details (e.g., how the goal-reward maximum or proxy target is instantiated per environment, and a concise summary of tuning budgets for compared off-policy baselines) would further strengthen reproducibility and make the evidence easier to audit.

**Requested Changes:**

Critical changes
1.	Please explicitly report, for each benchmark, the exact normalization constant or proxy target used in practice (including its numeric value and where it comes from, e.g., task configuration or reference-agent performance), and clarify how the computation is handled under early termination or variable episode lengths.
2.	For Figure 6(b) and 6(d), the curves do not clearly plateau by the end of the reported budget. Please clarify whether the intended takeaway is final performance or fixed-budget learning, and report budget-aware summaries (e.g., area under the learning curve and last-window averages). Extending training to show near-asymptotic behavior would further strengthen the claim if the paper emphasizes final performance.
Non-critical changes
3.	Provide a more systematic sensitivity analysis for the new method-specific hyperparameters (beyond the existing ablations), and summarize recommended defaults and common failure modes. This would strengthen the robustness claim and reduce the perceived tuning burden for practitioners.
4.	For completeness, briefly summarize the tuning search space/budget used for the off-policy baselines (e.g., how many configurations were tried, what ranges were used, and the selection criterion). The tuned hyperparameters are already reported; this addition would simply make the comparison easier to audit.
5.	Figure 6(a) does not display the timestep scale factor (for example, 1e8) that appears in other panels.
6.	Improve figure captions to explicitly state whether plotted curves correspond to total return or goal-related return, and remind the reader of the exact goal-achievement computation, since this quantity is central to the mechanism and claims.
7.	Add a brief practitioner-oriented note summarizing when the method is expected to be most beneficial (for example, dense-reward tasks with pronounced local optima) and any anticipated limitations (for example, unclear goal reward decomposition or unknown goal maxima), to improve adoption and reduce misuse.
8.	I also noticed a few minor typos/grammar issues (e.g., “incoporates” instead of “incorporates”), which can be easily fixed

---

> ### Author Response · Authors · 2026-01-20
> **Our response to reviwer PA8K**
>
> We thank the reviewer for the thoughtful and positive assessment of our work. We are encouraged that you found the motivation clear and the empirical results on HumanoidBench and IsaacLab aligned with our claims. Below, we outline how we have addressed your specific requests to strengthen the evidence and reproducibility of the manuscript.
>
> ### 1. Detailed Reporting of Benchmark Constants
>
> **Reviewer Concern**: Please explicitly report, for each benchmark, the exact normalization constant or proxy target used in practice (including its numeric value and where it comes from), and clarify how the computation is handled under early termination or variable episode lengths.
>
> **Response**:
> We add clarification and Table 6 to the Appendix, which lists the exact proxy targets $\hat{G}$ from the benchmark paper for every HumanoidBench task we used. For each task, we set the maximum training budget based on the difficulty level we observed from experience.
>
> ### 2. Extended Training
>
> **Reviewer Concern**: For Figure 6(b) and 6(d), the curves do not clearly plateau by the end of the reported budget.
>
> **Response**:
> Per your observation on Figure 6 (Figure 5 in the revised manuscript), we have extended the training budget for the Dribbling and Tightrope tasks to convergence. The updated plots now demonstrate that GAGE outperforms PPO in final performance.
>
> ### 3. Hyperparameter Sensitivity and Recommendations
>
> **Reviewer Concern**: Provide a more systematic sensitivity analysis for the new method-specific hyperparameters (beyond the existing ablations), and summarize recommended defaults and common failure modes.
>
> **Response**:
> We have included extra systematic sensitivity study in Appendix D.2, including Fig. 12, 13, and 14, covering method-specific parameters such as initial logit range $\delta_{z,0}$, logit factor $\alpha_\text{logit}$, initial std $\sigma_0$, and action mean factor $\alpha_\text{mean}$. Based on our experiments, we have added the “Best Practices” notes recommending starting values of hyperparameters.
>
> ### 4. Transparency in Tuning Budgets
>
> **Reviewer Concern**: For completeness, briefly summarize the tuning search space/budget used for the off-policy baselines.
>
> **Response**:
> We explicitly report that we first tuned SAC and MaxinfoSAC to get a strong baseline, as shown in Fig. 13 in Appendix. Regarding baseline fairness, we build on the public implementation and environments released with a published ICLR work. However, to reproduce the reported results and avoid relying on any unstated tuning effort, we additionally tuned SAC-v2 and MaxInfoSAC ourselves under the same evaluation protocol. For SAC-GAGE, method-specific hyperparameters were chosen using the same task set used in the MaxInfoSAC paper, and then reused unchanged across the remaining HumanoidBench tasks to avoid per-task overfitting.
>
> ### 5. Formatting and Clarifications
>
> We have corrected the missing scale factors in Fig. 6 (Fig. 5 in the new manuscript) and added an explanation for the “episode return” we report in the experiment plots. To be fair for baselines, we only report the “total return” including both the “goal reward” and “auxiliary rewards”. Only in Fig.2, we exclude the action penalty to isolate the influence of varying action penalty factors for all agents. We have fixed all identified typos.
>
> ### 6. Scope and limitations
>
> We have revised the manuscript to emphasize our focus on dense-reward tasks with pronounced local optima. We have extended our discussion on limitations and potential failure cases.

---

> > ### Comment · Reviewer_PA8K · 2026-02-04
> > **Response to the new revision**
> >
> > I thank the authors for their detailed response and the comprehensive revisions made to the manuscript. I appreciate that my requests regarding reproducibility and experimental validation have been thoroughly addressed.
> >
> > Specifically, the inclusion of the exact proxy targets in Table 6 and the extended training budgets in Figure 5 (formerly Figure 6) have resolved my concerns about the reproducibility of the benchmark settings and the asymptotic behavior of the agents. The extended curves now convincingly demonstrate the convergence properties on the Dribbling and Tightrope tasks.
> >
> > I also value the addition of Appendix D.2 and the "Best Practices" section, which offers much-needed transparency regarding hyperparameter sensitivity. This leads to one final observation/question: In the new sensitivity analysis for discrete PPO-GAGE (Figure 12), the method appears highly robust on tasks like Ant Acrobatics but shows significant sensitivity on the Balance Beam task—specifically, varying $δ_(z,0)$ from -5 to -3 seems to determine whether the task is solved or fails completely.
> >
> > Could the authors briefly clarify if this sensitivity is tied to the specific instability of the Balance Beam environment? While the recommended defaults are helpful, I am curious if practitioners should prioritize tuning $δ_(z,0)$ over other parameters when applying GAGE to similarly unstable tasks or environments with tight physical constraints (e.g., narrow safety margins).

---

> > > ### Author Response · Authors · 2026-02-04
> > > **Response to Reviewer PA8K**
> > >
> > > We thank the reviewer for their positive feedback regarding our revisions to the reproducibility and sensitivity analysis. We are pleased that Table 6 and the extended training curves in Figure 5 have resolved their concerns regarding benchmark settings and asymptotic convergence.
> > > Regarding the insightful question about the sensitivity of $\delta_{z,0}$ on the Balance Beam task in Figure 12:
> > >
> > > **Environmental instability**
> > >
> > > The reviewer is correct that this sensitivity is closely tied to the specific instability of the Balance Beam environment. Unlike Ant Acrobatics, where the search space for stable balance is relatively broad, the Balance Beam task involves extremely tight physical constraints, specifically a 0.1m narrow path and a tilted slope, where a single misstep can lead to immediate termination.
> > >
> > > The hyperparameter $\delta_{z,0}$ determines the initial "flatness" or probability lower bound of the discrete action distribution when goal achievement is zero. On the Balance Beam, a value of $\delta_{z,0} = -3$ corresponds to a higher lower bound on action probabilities. While this theoretically increases exploration, in this unstable task, it forces the agent to sample “dangerous” actions too frequently, leading to premature termination before a stable gait can be established. With $\delta_{z,0} = -5$, the agent begins with a slightly more concentrated action distribution that allows it to learn to avoid immediate falls, prioritizing “standing” while still maintaining sufficient exploration to discover subsequent gaits. As illustrated in the updated supplementary videos, learning proceeds through three distinct behavioral plateaus: first static balance, then a slow trot, and finally a 2m/s gallop.
> > >
> > > **Guidance for practitioners**
> > >
> > > Based on this observation, we agree with the reviewer’s assessment. For tasks with tight physical constraints or high sensitivity to initial conditions (unstable tasks), practitioners should prioritize tuning $\delta_{z,0}$. Ensuring the agent survives long enough to explore the environment's primary "bottleneck" is critical for successful learning. We update the "Best Practices" section (marked in blue) to explicitly state that for unstable environments, a more negative $\delta_{z,0}$ should be preferred to prevent early failure and enable the discovery of optimal behaviors.

---

### Review · Reviewer_nw2b · 2025-12-23

**Summary Of Contributions:**

This paper proposes Goal Achievement Guided Exploitation (GAGE), a method for regulating exploration in reinforcement learning by enforcing adaptive lower bounds on policy stochasticity based on a measure of goal achievement. Instead of adding entropy regularization to the reward, GAGE directly constrains the policy distribution—via the standard deviation in continuous action spaces and the logit range in discrete ones—ensuring that exploration does not vanish until the agent approaches its target performance.
The method is integrated into both on-policy (PPO) and off-policy (SAC) algorithms and evaluated on a suite of challenging dense-reward robotic control tasks, including custom IsaacLab environments and HumanoidBench. The experiments demonstrate that GAGE improves robustness and learning stability compared to PPO baselines and avoids certain forms of premature convergence observed with entropy-based exploration.
Key strengths include a clear critique of entropy regularization, an interpretable goal-performance-based exploration mechanism, and extensive experiments on difficult control tasks.
Key weaknesses are insufficient comparison to strong maximum-entropy and exploration-scheduling baselines, and an overstatement of conceptual novelty relative to what is effectively a performance-based exploration scheduling method.

**Audience:**

Yes

**Audience Explanation:**

The paper addresses an important and underexplored issue—premature convergence in dense-reward continuous control—and proposes a practical, interpretable mechanism that many researchers in reinforcement learning and robotics would find interesting. The empirical results on challenging humanoid and locomotion tasks, as well as the critique of entropy-based exploration, are likely to be informative to a subset of the TMLR audience.

However, interest alone does not outweigh the current limitations in positioning and evaluation.

**Broader Impact Concerns:**

The submission does not raise immediate or severe ethical concerns. The proposed method focuses on improving exploration strategies in reinforcement learning and is evaluated primarily in simulated robotic control environments.

**Claims And Evidence:**

No

**Claims Explanation:**

While the experimental results convincingly show that GAGE outperforms PPO and avoids premature convergence in the selected tasks, the paper’s central claims go beyond what the evidence supports.
The work is framed as a fundamental alternative to maximum entropy reinforcement learning, yet:

- First, the paper positions GAGE as an alternative to maximum entropy reinforcement learning, yet the strongest empirical gains are shown against PPO, which is not a principled MaxEnt RL algorithm. Comparisons against SAC and MaxInfoSAC—the most relevant MaxEnt-based baselines—show similar final performance, with improvements largely limited to learning speed.

- Second, the method effectively implements adaptive variance/entropy scheduling, but is not compared against other scheduling strategies (manual annealing, performance-based schedules, KL-constrained methods, or learned schedules).

- Third, the paper criticizes newer entropy-based methods (e.g., SAC-v2) on the grounds that entropy tuning is difficult, yet does not evaluate SAC-v2 empirically, while GAGE itself introduces several hyperparameters that require tuning. This asymmetry undermines the argument that GAGE meaningfully reduces tuning burden relative to modern MaxEnt approaches.

- Fourth, although the paper discusses intrinsic-reward–based exploration methods, it uses Random Network Distillation (RND) as the representative baseline. RND is explicitly designed for sparse-reward exploration, and its poor performance in dense-reward robotic control tasks is well known. Using RND in this setting does not provide a meaningful comparison and weakens claims about outperforming intrinsic motivation methods.

- Finally, a key assumption of the method is access to a well-defined goal reward or a reliable approximation of the optimal goal return. In many real-world or complex RL tasks, separating goal rewards from auxiliary shaping terms—or estimating a meaningful proxy for the optimal return—is non-trivial. While the paper explores heuristic approximations, the sensitivity of GAGE to mis-specified or poorly estimated goal achievement is not thoroughly analyzed. This requirement limits the general applicability of the method beyond carefully designed dense-reward robotic control settings.

As a result, the experimental evidence does not fully substantiate the paper’s stronger claims about replacing or fundamentally improving upon maximum entropy RL.

**Requested Changes:**

Below is a list of proposed adjustments, indicating which are critical for acceptance and which would strengthen the paper.

# Critical Changes (Required for Acceptance)
1. Stronger and more appropriate baseline comparisons
The method should be compared against:
- entropy annealing schedules,
- performance-based entropy or variance schedules,
- KL-constrained policy optimization methods (e.g., MPO-style constraints),
- and other adaptive exploration mechanisms beyond PPO-style entropy bonuses.
Without these comparisons, it is not possible to assess whether GAGE offers advantages beyond well-designed scheduling heuristics.

2. Clearer positioning of GAGE as an exploration scheduling method
The paper should either:
- substantially strengthen evidence that GAGE outperforms strong MaxEnt methods (e.g., SAC variants) in final performance, tuning difficulties with a fair comparison, or
- explicitly reframe GAGE as a principled exploration scheduling strategy rather than a replacement for maximum entropy RL.

3. More cautious claims regarding MaxEnt RL
The critique of maximum entropy RL should be moderated to reflect that modern methods (e.g., SAC-v2) already address some of the stated issues pragmatically. Claims should be aligned more closely with empirical findings.

4. Reconsideration of intrinsic-reward baselines
Using RND as the primary intrinsic-reward baseline in dense-reward continuous control tasks is not appropriate.
Either more suitable intrinsic exploration methods should be evaluated, or claims regarding intrinsic-reward approaches should be substantially toned down.

# Non-Critical but Strengthening Changes
5. Sensitivity analysis of GAGE hyperparameters
A more systematic study of sensitivity to σ₀, goal smoothing, and proxy goal estimation would strengthen claims of robustness and ease of tuning.

6. Clarification of scope and limitations
The paper should more explicitly state that GAGE is designed for dense-reward, single-goal tasks and discuss how it might (or might not) extend to sparse, multi-goal, or stochastic environments.

7. Additional theoretical discussion
While full convergence guarantees are not required, discussion of potential failure modes (e.g., over-exploration, misestimated goal achievement) would improve completeness.

8. Clarification of scope
The paper should more explicitly limit its claims to dense-reward, single-goal robotic control tasks, and discuss potential failure modes outside this regime.

---

> ### Author Response · Authors · 2026-01-20
> **Our Response to Reviewer nw2b - Part 1**
>
> We thank the reviewer for their rigorous and insightful critique. We appreciate the recognition that our work addresses an "important and underexplored issue" (premature convergence in dense-reward control). We accept the reviewer’s feedback regarding the framing of our method and revise the manuscript to focus on GAGE as a principled performance-guided exploration scheduling strategy.
> ### 1. Stronger and More Appropriate Baseline Comparisons
> **Reviewer Concern**: PPO is not a principled MaxEnt algorithm and that SAC/MaxInfoSAC show similar final performance and it is requested to have comparisons against more methods.
>
> **Response**:
> - Manual vs. Adaptive Schedules: We agree that GAGE can be viewed as an automated schedule. In response, we have included new results in Fig. 9 from the Appendix comparing GAGE against several manually designed schedules. At the same time, we want to point out that the superior performance of GAGE comes from not only the prior-based adaptive schedule, but also the employment of more interpretable and controllable stochasticity measures instead of entropy. We also provide the result of PPO with manually designed schedules for policy standard deviation $\sigma$. The ablation study shows that both the prior knowledge of goal achievement and the new stochasticity measures boost exploration performance.
> - MPO-style constraints: As far as we know, the MPO authors have not published their implementations. It is difficult to do a fair comparison. We would try to add it in the future revisions if the work is open sourced. It would also be well appreciated, if the reviewer can guide us to a proper publicly available implementation.
> - Beyond PPO: The SAC and MaxinfoSAC baselines are implemented with the more advanced version SAC-v2. We have corrected the baselines’ names accordingly. Additionally, we provide new results comparing GAGE, tuned SAC-v2, and tuned MaxinfoSAC on 9 more tasks from the HumanoidBench benchmark in Sec. 4.1. GAGE achieves better final performance or faster learning speed across the many of the compared tasks.
> ### 2. Reframing and Positioning
> **Reviewer Concern**: GAGE should be framed as a scheduling method rather than a foundational replacement for MaxEnt RL.
>
> **Response**:
> - Revised Positioning: We change our focus from MaxEnt RL to entropy-based exploration, since PPO is not regarded as a principled MaxEnt RL algorithm. We acknowledge that GAGE, analogous to MaxEnt, promotes policy stochasticity during training. Thus, GAGE does not replace, but complements the MaxEnt framework. To properly position our motivation and contributions, we explicitly reframe GAGE as a principled, performance-based exploration scheduling framework that provides a robust alternative to standard entropy bonuses.
> - The “rethinking” aspect: Our "rethinking" of entropy-based exploration stems from the observation its following application hurdles:
>     * Objective mismatch: Using entropy as a soft regularizer in the optimization objective can shift the optimal solution and requires manual temperature annealing to recover the original objective. In contrast, by directly regulating the distribution parameters, like standard deviation $\sigma$ and logit range $\delta_z$, GAGE preserves the original objective during the whole training period.
>     * Non-trivial exploration scheduling: It is difficult to schedule exploration for entropy-based methods, especially for tasks requiring different exploration at different learning phases. By incorporating the prior knowledge of goal achievement, GAGE realizes adaptive exploration scheduling. The comparison can be found in Fig. 9 from the Appendix. We further add discussion about the design choice between entropy coefficient and policy std, suggesting that directly changing the policy std could potentially work better compared to scheduling the entropy coefficient in the continuous control tasks.
>     * Soft constraints: While entropy-based methods encourage stochasticity, it cannot theoretically guarantee a lower bound on action probabilities. GAGE introduces a hard lower bound on action probabilities. To show the effect, we add a new paragraph, “Non-vanishing exploration”, in Sec. 4.2 to analyze the action’s probabilities at different learning phases. It clearly shows how multiple actions of the baseline agents collapse to zero, while GAGE lower-bounding them according to the goal achievement.
> (To be continued in the next part)

---

> ### Author Response · Authors · 2026-01-20
> **Our Response to Reviewer nw2b - Part 2**
>
> ### 3. Moderating Claims Regarding Modern MaxEnt
> **Reviewer Concern**: Modern methods like SAC-v2 already address some tuning issues pragmatically.
>
> **Response**:
> We acknowledge the pragmatic successes of SAC-v2 in our manuscript. And we moderate our claims  However, we maintain that SAC-v2 targets a fixed entropy, whereas GAGE dynamically adjusts exploration based on actual task progress (goal achievement). This allows the policy to converge to a deterministic state naturally, whereas SAC often maintains a fixed level of noise unless the temperature is manually annealed.
>
> ### 4. Reconsideration of Intrinsic Reward Baselines (RND)
> **Reviewer Concern**: RND is designed for sparse rewards; its poor performance in dense-reward tasks is expected and does not validate GAGE's superiority.
>
> **Response**:
> - Clarifying the Choice of RND: We included RND to test if curiosity-driven methods could alleviate premature convergence in dense-reward settings. We concede that RND's novelty-seeking can be counterproductive in high-dimensional control where most "novel" states are failed trajectories (e.g., falling down).
> - Updated Discussion: We substantially tone down claims of "outperforming intrinsic motivation" and instead frame the discussion around the different focus of these methods: novelty-seeking for sparse rewards vs. performance-guided exploitation for dense-reward local optima.
>
> ### 5. Sensitivity Analysis and Limitations (Strengthening Changes)
> **Reviewer Concern**: Sensitivity analysis of hyperparameters ($\sigma_0$, goal smoothing) and the difficulty of estimating goal rewards.
>
> **Response**:
> - Hyperparameter Robustness: Analogous to PPO-GAGE, we have included an ablation on hyperparameters for PPO-GAGE with discrete action space and SAC-GAGE in the Appendix and provide advice to tune the hyperparameters.
> - Proxy Goal Estimation: We provide additional experiments and discussion for a larger range, ${0.1,0.25,0.5,1,1.5,2,5,10\}\times$, of the estimated total reward goal $\hat{G}$, investigating the effect of overly large/small target goal.
>
> ### 6. Clarification of scope, discussion of limitations and potential failure modes
> **Reviewer Concern**: Explicitly limit its claims to dense-reward, single-goal robotic control tasks, and discuss potential failure modes outside this regime.
>
> **Response**:
> We have revised the manuscript accordingly. Besides the clarified scope, we extend the discussion for limitations and potential failure modes in the last chapter.

---

> ### Author Response · Authors · 2026-02-02
> **Follow-up: Addressing Feedback from Reviewer nw2b**
>
> Dear Reviewer nw2b,
>
> We would like to kindly follow up on our response and the revised manuscript submitted 13 days ago.
>
> Your initial feedback provided crucial directions for improving our work. In response, we have performed substantial revisions, including adding new experiments and reframing the theoretical framework, specifically to address your concerns.
>
> As the discussion period continues, we are eager to hear your thoughts on whether these updates adequately resolve the points you raised. We remain ready to provide any further clarifications or conduct additional analysis if necessary.
>
> Thank you for your time and for the detailed guidance you have provided thus far.

---

> > ### Comment · Reviewer_nw2b · 2026-02-02
> > **Response to the new revision**
> >
> > I appreciate the authors’ efforts in the revision. Some of my main concerns have been addressed—most notably, the paper is now positioned more clearly as an exploration scheduling method rather than an improvement over SAC, and the added baseline comparisons are a step in the right direction.
> >
> > Overall, the revised claims and scope are moving in the right direction. That said, there are still instances of overclaiming. For example, the abstract states: “GAGE improves learning stability and final performance over various strong baselines for both on-policy and off-policy algorithms by a clear margin.” This is not what Figure 4 shows. Relative to SAC-v2 or MaxInfoSAC, there is little to no improvement in final performance in most environments, and any gains appear primarily in learning speed or stability. This point matters because the paper’s central framing is that GAGE is an alternative to maximum-entropy RL methods; therefore, performance against SAC-style baselines is among the most important evidence for the paper’s positioning.
> >
> > One additional question: in Figure 5, GAGE appears to progress through distinct learning stages, with sudden jumps in performance (especially in subplots (c) and (e)). How should these abrupt transitions be interpreted? Are they an artifact of the scheduling mechanism (e.g., crossing a goal-achievement threshold), or do they reflect some other underlying change in the policy or exploration regime?

---

> > > ### Author Response · Authors · 2026-02-03
> > > **Response to reviewer's new feedback**
> > >
> > > We thank the reviewer for their continued engagement and for recognizing that our revised positioning and baseline comparisons are moving in the right direction. We address your two remaining concerns below.
> > >
> > > ### Addressing "Overclaiming" in Final Performance
> > >
> > > We appreciate the reviewer’s nuanced feedback regarding our Abstract's phrasing and the interpretation of Figure 4. While GAGE demonstrates advantages on the HumanoidBench suite, we acknowledge that for several of these tasks, the final performance is comparable to well-tuned SAC-v2 and MaxInfoSAC.
> > >
> > > - **Revision**: We revise the abstract to state that GAGE "improves learning efficiency and stability across various strong baselines, achieving competitive or superior final performance" to better match the evidence found in Figure. 4. The revised manuscript has been uploaded. The revised abstract text is marked in blue.
> > >
> > > - **Positioning**: Following the reviewer’s earlier recommendation, we refocus the paper from critiquing the “maximum entropy” paradigm to studying entropy regularization as a widely used exploration heuristic. Under this framing, PPO and SAC-v2 serve as representative entropy-regularized baselines, and GAGE provides a more interpretable alternative to entropy-bonus tuning regardless of the specific backbone.
> > >
> > >
> > > ### Interpreting "Sudden Jumps" in Figure 5
> > >
> > > We appreciate the reviewer’s insightful observation regarding the abrupt performance transitions in the experiments (particularly subplots (c) and (e) from Figure. 5). We clarify that these “jumps” are characteristic of specific tasks and are further facilitated by GAGE’s performance-conditioned scheduling. Tasks such as Cartwheel and Dog Balance Beam contain multiple behaviour modes, where progress often occurs in stages.
> > >
> > > As an example, training on Dog Balance Beam typically follows a staged learning process; we provide videos from actual checkpoints illustrating the three corresponding gaits in the supplementary material.
> > > - **Stage 1 (Standing)**: With near-zero goal achievement, the robot focuses on the primary constraint—maintaining balance and standing on the beam.
> > > - **Stage 2 (Trotting)**: The robot identifies a stable trotting gait. As goal achievement improves, GAGE begins to lower the exploration bound, leading to a performance spike.
> > > - **Stage 3 (Galloping)**: The robot transitions to a high-speed running gait. Once this optimal motion is identified, the policy quickly converges, resulting in the final "jump" to maximum returns.
> > >
> > > Overall, we believe these transitions reflect the policy navigating between local optima in tasks with different behaviour modes. The observed “jump” behavior is partially attributable to GAGE’s scheduling mechanism: by maintaining exploration early (via the hard bound) and then relaxing the lower bound on the policy logit range as goal achievement increases, GAGE provides “patience” to search for better gaits and enables faster exploitation once a superior mode is found.

---

> > > > ### Comment · Reviewer_nw2b · 2026-02-04
> > > >
> > > > I thank the authors for the update and the explanation. I believe the overclaiming in the abstract has been addressed. The explanation of the sudden performance jumps also makes sense to me.
> > > >
> > > > I have no further questions at this point.

---

### Review · Reviewer_7HFz · 2026-01-06

**Summary Of Contributions:**

This paper presents an adaptive exploration method in reinforcement learning (RL). The authors propose Goal Achievement Guided Exploitation (GAGE), a framework that regulates exploration based on the agent’s goal achievement. Starting from the classical maximum entropy (MaxEnt) principle, the paper highlights its implicit and ambiguous nature regarding exploration. This leads the authors to define a new objective for goal achievement, $g(\pi) = J_{\mathrm{g},\pi} / J_{\mathrm{g},\pi_\ast}$ based on auxiliary reward function $r_\mathrm{g}$. Using a variant of measures defined via $g(\pi)$, the GAGE method successfully guides agent policies to accomplish discrete and continuous control tasks, including those with very long behavioral sequences. The paper provides extensive empirical studies demonstrating the effectiveness of GAGE.

**Additional Comments:**

The symbol g in $g(\pi)$ and $J_{\mathrm{g},\pi}$ are used in similar contexts but with slightly different meanings. Please use distinct symbols to clearly distinguish each concept to avoid confusion.

**Audience:**

Yes

**Audience Explanation:**

Goal-oriented RL is one of the most practical methods for enabling agents to learn sophisticated action sequences. The general performance benefits demonstrated by GAGE are impressive, and I find the experiments were well-conducted.

**Claims And Evidence:**

No

**Claims Explanation:**

While I do think this can be a promising direction regarding reward engineering in goal-oriented RL with good performance, I also find its claims regarding MaxEnt RL to be puzzling or ambiguous.
* I do not particularly agree with its main claim, the abstract notion that the maximum entropy does not solve (or cause) premature convergence. Theoretically, maximum entropy aids RL optimization by encouraging exploration while guaranteeing the uniqueness of the solution. Empirically, RL algorithms often get stuck in local minima due to realistic constraints, such as the vastness of the solution space and the instability of estimating moving targets.  In other worlds, goal-oriented approaches like GAGE likely succeed in practice not because they rectify a fundamental limitation of MaxEnt, but because they inject useful, stable prior knowledge (e.g., enforcing a humanoid robot to proceed with specific velocities). I believe the authors mistakenly frame MaxEnt RL as having a fundamental, universally agreed-upon flaw in terms of exploration without providing convincing theoretical consideration for such a limitation.
* I do not understand the implication of example Fig 1, which is used to criticize the entropy measure. It may be true that two distributions have the same entropy; entropy is an indirect signal for exploration, whereas exploitation of returns takes precedence. The combination of reward and entropy typically balances this to ensure sufficient exploration. I totally agree $r_g$ is far more direct and stable, but this is not because it acts as a universally superior approach to exploration. Rather, it simplifies the original problem by providing guidelines for exploitation drawn from prior knowledge. I encourage the authors to clearly specify exactly which concept they are criticizing regarding the theoretical or empirical expectations of entropy-based exploration.
* I do not understand the term flatness throughout the paper. Are there citations for this specific definition? I encourage the authors to provide a detailed explanation of how flatness helps readers evaluate whether an action distribution is good. I also find it difficult to digest the concept of “sufficient exploration” as presented, specifically why establishing lower bounds on action probability is linked to such sufficiency.

**Requested Changes:**

* Please clarify exactly what issue GAGE attempts to solve regarding maximum entropy reinforcement learning.
* I suggest the authors focus more on the identification of useful priors in robot reinforcement learning and provide a generalization of their findings that aims solely at exploitation (or goal-guidance) rather than framing it as a fix for entropy-based exploration.
* Provide either theoretical arguments or convincing evidence (e.g., toy experiments, citations) to support the central claim that MaxEnt causes the specific convergence issues described.

---

> ### Author Response · Authors · 2026-01-20
> **Our response to reviwer 7HFz**
>
> We thank the reviewer for the constructive feedback and for recognizing the performance benefits and well-conducted experiments of GAGE. We address the specific concerns regarding Maximum Entropy (MaxEnt) RL, the definition of “flatness”, and the interpretation of our results below.
>
> ### 1. Clarification of MaxEnt RL limitations vs. GAGE
>
> **Reviewer Concern**: MaxEnt aids optimization and that GAGE succeeds by injecting prior knowledge rather than fixing a fundamental flaw in MaxEnt.
>
> **Response**:
> We acknowledge that GAGE, analogous to MaxEnt, promotes policy stochasticity during training. Thus, GAGE does not replace, but complement the MaxEnt framework. To properly position our motivation and contributions, we have revised the paper, mainly the abstract, introduction, background, and discussion.
> We do not claim MaxEnt is universally flawed, but rather that existing RL algorithms face the following practical hurdles due to entropy-based exploration:
> * Objective mismatch: Using entropy as a soft regularizer in the optimization objective can shift the optimal solution and requires manual temperature annealing to recover the original objective. In contrast, by directly regulating the distribution parameters, like standard deviation $\sigma$ and logit range $\delta_z$, GAGE preserves the original objective during the whole training period.
> * Non-trivial exploration scheduling: It is difficult to schedule exploration ratio using entropy, especially for tasks requiring different exploration at different learning phases. By incorporating the prior knowledge of goal achievement, GAGE realizes adaptive exploration scheduling.
> * Soft constraints: While entropy-based methods encourage stochasticity, it cannot theoretically guarantee a lower bound on action probabilities. GAGE introduces a hard lower bound on action probabilities. To clearly support the effect, we add a new paragraph, “Non-vanishing exploration”, in Sec. 4.2 to analyze the action’s probabilities at different learning phases. It clearly shows how multiple actions of the baseline agents collapse to zero, while GAGE lower-bounding them according to the goal achievement.
>
> ### 2. Interpretation of example figure 1 and the usage of entropy for distribution stochasticity
>
> **Reviewer Concern**: The combination of reward and entropy typically balances this to ensure sufficient exploration”, and that $r_g$ “a universally superior approach to exploration” than entropy-based exploration.
>
> **Response**:
> As far as we know, there is no formal definition of “sufficient exploration”. To avoid misunderstanding, we replace “sufficient exploration” with “non-vanishing exploration”, which can be indicated by the minimum action probability guaranteed by GAGE.
> As we show in the additional data analysis in Fig. 7 of Sec. 4.2, GAGE ensures lower bounds for action probabilities, which can not be guaranteed by entropy regularization. The entropy-based agents collapse the probabilities of the proper actions prematurely to below 1e-10, even when they have not learned to move at all. As a result, they will almost not explore these actions during the following training, despite a relatively high entropy maintained by several improper actions. Furthermore, as the reviewer also pointed out, entropy is an indirect signal for exploration, making the tuning of the entropy coefficient very difficult. We provide additional experiment results comparing different schedules of entropy coefficient and standard deviation $\sigma$, showing that the latter one is much more stable and easier to tune.
>
> ### 3. Interpretation of "Flatness"
>
> **Reviewer Concern**: The term "flatness" is difficult to digest.
>
> **Response**:
> We replaced “flatness” with “stochasticity” before formally defining and explaining it in Sec. 3.2. “Flatness” $\mathcal{F}(\pi)$ is not a formally defined value, but a generic concept we introduced to measure how “spread out” a distribution is. Its exact definition can vary for different distributions. Specifically:
> * Standard deviation $\sigma$ for Gaussian distribution
> * Negative logit range $\delta_z$ for categorical distribution
>
> ### 4. Identification of priors
>
> Following the reviewer’s suggestion, we revise our introduction and method chapters to highlight that GAGE uses performance-based priors to guide exploration. At the same time, we argue that GAGE also provides a solution to entropy regularization’s practical hurdles: *objective mismatch* and *soft constraint*.
>
> ### 5. Clarify the symbols $g(\pi)$ and $J_{\text{g},\pi}$
>
> We acknowledge the confusion. In the revised manuscript, we use $J_{\text{goal},\pi}$ to avoid any ambiguity.

---

> ### Comment · Reviewer_7HFz · 2026-01-30
> **A follow-up question**
>
> I thank the authors for the response. I have a further question. I believe the authors introduced a new terminology "stochasticity", defined as
> $$ \texttt{f}(g(\pi))$$
> where the function $\texttt{f}$ is an affine function? If we closely look into the definition of $g$, it is not specifically measuring quantities related to distributional properties involving $\pi$. Instead, it is defined as sum of $r_{goal}$ which is likely to be a problem-specific values determined by samples from $\pi$. How could the authors justify the naming "stochasticity" in the general information theoretical context? How could the idea of stochasticity holds in other subfields of machine learning?

---

> > ### Author Response · Authors · 2026-01-30
> > **Response to the follow-up question**
> >
> > Thank you for your follow-up and for the opportunity to clarify these definitions.
> >
> > Your understanding of the goal achievement ratio $g$ is correct: it measures the task-specific performance ratio of a given policy $\pi$. However, we would like to clarify the distinct roles of the function $f(g(\pi))$ and the term "stochasticity."
> >
> > **Clarification of the Scheduling Framework**
> >
> > We do not define "stochasticity" itself as $f(g(\pi))$. Instead, as detailed in Section 3.2, we formalize a generic measure of distributional stochasticity termed "flatness", denoted as $\mathcal{F}(\pi)$. The function $f(g(\pi))$ serves as a customizable scheduling function that defines the adaptive lower bound for this flatness.
> >
> > This framework allows GAGE to instantiate specific constraints depending on the action space:
> > - Continuous Policies: $f(g(\pi))$ determines the minimum standard deviation $\sigma_{LB}$.
> > - Discrete Policies: $f(g(\pi))$ determines the minimum logit range $\delta_{z,LB}$.
> >
> > Concretely, $f$ maps the current performance level to a required lower bound on the policy's spread. This mechanism explicitly regulates how exploration decays as the agent approaches optimal performance, effectively automating the exploration-exploitation trade-off based on real-time performance feedback rather than a rigid, time-based schedule.
> >
> > **On the Term "Stochasticity"**
> >
> > We acknowledge that "stochasticity" can be interpreted broadly across different fields. However, within Reinforcement Learning (RL) literatur, particularly regarding entropy-regularized framework, this term is conventionally used to denote the degree of randomness in the action distribution. Our naming convention aligns with established works such as [1], [2], and [3], which leverage policy entropy as proxy for exploratory behavior.
> >
> > [1] Haarnoja, T., et al. (2018). Soft Actor-Critic: Off-Policy Maximum Entropy Deep Reinforcement Learning with a Stochastic Actor. ICML.
> >
> > [2] Eysenbach, B., & Levine, S. (2022). Maximum Entropy RL (Provably) Solves Some Robust RL Problems. ICLR.
> >
> > [3] Ahmed, Z., et al. (2019). Understanding the Impact of Entropy on Policy Optimization. ICML

---

> ### Comment · Reviewer_7HFz · 2026-01-30
>
> I thank the authors for the clarification. I think exposing the setups of $\mathcal{F}$ explicitly close to the definition of $\mathcal{F}_\mathrm{LB}$ would benefits the presentation of Sec. 3.2. I do not have further questions at this moment.

---

> > ### Author Response · Authors · 2026-02-03
> > **Response to Reviewer 7HFz**
> >
> > We thank the reviewer for their final positive feedback and for the helpful suggestion to improve the presentation of Section 3.2. Following their recommendation, we have revised the manuscript (with changes marked in blue) to explicitly expose the setups of our flatness measure $\mathcal{F}$ close to the formal definition of the adaptive lower bound $\mathcal{F}_{LB}$. We appreciate the time and effort the reviewer have invested in reviewing our work.

---

### Author Response · Authors · 2026-01-20
**Summary of Changes**

We thank the reviewers for their constructive feedback. In response, we substantially revised the manuscript to (i) align the framing and claims with what the evidence supports, (ii) strengthen baseline comparisons, especially against SAC-v2, entropy-coefficient scheduling and automated tuning approaches—and (iii) improve clarity, reproducibility, and practitioner guidance. Our changes are highlighted in red for review purposes.
### 1. Repositioning and moderated claims about MaxEnt RL
   * We reframed GAGE as a **performance-based exploration scheduling / distribution-constraint framework**, rather than a general “fix” or replacement for maximum-entropy RL.
   * We softened and clarified the MaxEnt discussion: entropy regularization is an **indirect control knob** that optimizes a **regularized objective by design**; our focus is on practical issues in dense-reward tasks (premature collapse, schedule sensitivity, profound local optima), not a universal theoretical flaw.
   * We scoped our claims more explicitly to **dense-reward, goal-directed robotic control** where a meaningful goal-return target (or proxy) is available.
   * We added an additional analysis for discrete action distribution obtained by GAGE and PPO to show the premature convergence issue intuitively.
### 2. Title, abstract, and introduction rewritten for accuracy and scope
   * We revised the **title/abstract/introduction** to reflect the updated positioning, reduce overclaiming, and clearly state scope/assumptions.
### 3. Stronger baselines and clearer comparisons
   * For PPO, we expanded comparisons to include:
     * Manually designed entropy-coefficient schedules
     * **Population-Based Training (PBT)** [1] for automated entropy tuning
   * We toned down broad claims about intrinsic-reward methods and clarified that **RND** is primarily designed for sparse-reward exploration and is included as a reference baseline.
### 4. Direct experimental justification: scheduling std vs. scheduling entropy coefficient
   * We added a controlled ablation applying the same schedule families (**Linear Decay**, **Cosine Decay**, **Cosine Annealing with Warmstarting**) to either the entropy coefficient or the policy standard deviation, while keeping other PPO settings fixed.
   * We show a clear asymmetry: **entropy-coefficient schedules often underperform the best fixed tuned entropy coefficient** in our setting, whereas **std schedules are meaningfully competitive** (cosine std decay can match GAGE on some tasks).
   * Overall, **GAGE performs best across the suite** without per-task schedule selection and adaptively learns strong schedules.

### 5. Off-policy evaluation strengthened and expanded
   * We expanded the HumanoidBench evaluation to **12 tasks**.
### 6. Reproducibility: explicit Ĝ reporting and sensitivity analysis
   * We clarified how the proxy goal **Ĝ** is set in practice (including HumanoidBench target-return usage).
   * We added a broader sensitivity study with:
     * **Ĝ ∈ {0.1, 0.25, 0.5, 1, 1.5, 2, 5, 10} ×** reference provided by tuned PPO runs
   * We added a concise limitation/practitioner note showing that overly large **Ĝ** can keep goal achievement low, maintain excessive exploration, and hinder convergence, while using too small **Ĝ** could cause premature convergence, leading to low performance.
### 7. Hyperparameter analysis and practitioner guidance
   * We added/expanded sensitivity studies and recommended defaults for **continuous/discrete PPO-GAGE** and **SAC-GAGE**.
### 8. Clarity improvements and minor fixes
   * We addressed notation ambiguity (e.g., symbol overload), improved figure captions (what return is plotted / how goal achievement is computed), and fixed typos and phrasing issues throughout.


We are happy to clarify any remaining questions.

[1] Jaderberg, Max et al. “Population Based Training of Neural Networks.” ArXiv abs/1711.09846 (2017): n. pag.

---

### Author Response · Authors · 2026-01-30
**Gentle Reminder: Request for Feedback on Revised Manuscript**

Dear Reviewers,

We would like to follow up on the revised manuscript and the point-by-point responses we submitted ten days ago.

We have worked diligently to address the concerns raised during the first round of reviews. We would greatly appreciate your thoughts on these updates to ensure our revisions are moving in the right direction. We remain open to further suggestions to continue improving the work.

Thank you again for your time and for your dedication to the peer-review process.

---

### Decision · Action_Editor_BVr1 · 2026-02-17

**Recommendation:** Accept with minor revision

**Additional Comments:**

I am on the fence between rejecting the paper with encouragement to resubmit and accepting the paper after minor revisions. However, as the necessary revisions can be easily verified by myself, I decided for the latter options to reduce burden on the reviewers. However, I do note that __the camera-ready version will not be automatically accepted__, and I urge the authors to carefully address the following issues and, more generally, __to ensure a more objective presentation__.

Claims & Evidence
-----------------
The submission still contains several claims that are not supported by evidence, and that often appear factually wrong:

__Claim__: "Unlike entropy regularization GAGE preserves the original objective."
"Because GAGE does not modify the reward function, it preserves the original learning objective"

The fact that GAGE does not modify the reward function is not sufficient to deduce that GAGE preserves the learning objective. Indeed, by constraining the search space, GAGE may end up with different solutions. While the flatness lower bound may diminish during learning, the same can be achieved with entropy regularization, where the coefficient can be decreased to zero (e.g., for entropy-regularized PPO).

__Claim__: "It is not easy to implement hard constraints directly on [entropy]"

The submission does not provide any evidence for this claim. The conditional entropy of a Gaussian policy can be computed in closed form and is differentiable. It would be straightforward to implement a differentiable projection layer that scales a covariance matrix to exactly achieve a desired entropy. The fact that there doesn't seem to be much interest in such precise control of the entropy, is not sufficient to claim that it would be hard to do so. If we are talking about the expected entropy to come (as in MaxEnt-RL), I agree that it is not eays to implement hard constraints. However, this is a consequence of sequential decision making; imposing hard constraints on the expected flatness to come, would be similarly hard.

__Claim__: "[entropy coefficient schedules] are difficult to design and sensitive to reward scaling"

I do not see why entropy schedules need to be sensitive to reward scaling. For example Abdolmaleki et al. (Model-based relative entropy policy search, NeurIPS 2015) proposed to initially lower bound the entropy to the entropy of the initial policy and exponentially decayed this bound. Instead, it seems like the difficulty of specifying appropriate schedules is alleviated in GAGE by assuming that the optimal task reward is approximately known. Such knowledge is typically not assumed and could be exploited also for adaptive entropy schedulers.

__Claim__: "directly constraining $\sigma$ is more intuitive and numerically stable [compared to entropy]"

It is not clear to me why constraining $\sigma$ is numerically more stable, since there is a one-to-one mapping between the entropy and standard deviations of a Gaussian distribution.

__Claim__: "entropy acts as a lossy summary statistic: multiple distributions can share identical entropy values while having vastly different minimum action probabilities"

This is correct. However, we can similarly argue that the minimum action probability is a lossy summary statistics, and that different policies with the same minimum action probability may have vastly different exploration efficiency.

Terminology
-----------
The submission introduces the term "flatness" which is okay, but a bit overloaded. However, the submission does not provide a general definition, but defines it per example (For Gaussians the flatness corresponds to the standard deviation, for discrete distributions it corresponds to the negative logit spread), which may be confusing to the reader. It would be better to clarify, that flatness is not a precise terminology, but that we attempt to capture action space coverage and select these two quantities, because larger values correspond to better coverage.

Providing definitions for the lower bounds (Eq.2 and Eq.5) is also a bit misleading. It would be better to state that the lower bounds are heuristically chosen as an affine function of the goal achievement.

**Audience:**

Yes

**Audience Explanation:**

The paper proposes to encourage exploration by enforcing lower bounds on probabilities (in the discrete setting) or on the variance (in the continuous setting with Gaussian policies). These bounds are automatically adapted during reinforcement learning based on the ratio between the current task reward (ignoring auxiliary terms for shaping) and a target task reward (which is assumed to be given). As exploration is an important problem in RL, and the proposed method shows some empirical success, I argue that at least some individuals in TMLR's audience would be interested.

**Claims And Evidence:**

Yes

**Claims Explanation:**

The initial submission suffered from over-claiming and unsubstantiated claims about entropy regularization, which was also noted by reviewer 7HFz and nw2b. During the rebuttal the submission was significantly improved by toning down, and now two reviewers support acceptance. Reviewer 7HFz still believes that the submission does not provide sufficient evidence and, furthermore, needs to introduce and justify the terminology better. I agree that the submission still lacks evidence to support some of its claims, and that the definitions for flatness and its lower bound are a bit confusing.

Hence, while I selected "yes" to the question above, it should be regarded as a "conditional yes", assuming that the required revisions are made (see "Additional Comments").

---

> ### Author Response · Authors · 2026-03-12
> **Response to action editor**
>
> Dear Action Editor,
>
> Thank you for your thoughtful and detailed feedback during the revision process. We have carefully addressed all of your concerns in the camera-ready version. Below, we summarize the changes made in response to each of your point.
>
> 1. "Unlike entropy regularization GAGE preserves the original objective." / "Because GAGE does not modify the reward function, it preserves the original learning objective"
>
> We agree that this claim was overstated. Because of the imposed constraint, GAGE also modifies the effective optimization problem and only recovers the original objective as the lower bound vanishes. We have removed the relevant parts from the Abstract, the Introduction, Section 3.3 (Advantages of GAGE) and Section 5 (Discussion).
>
> 2. "It is not easy to implement hard constraints directly on [entropy]"
>
> We appreciate this technical correction. We have removed the claim that such constraints are "not easy" to implement from Section 2.1. Instead, we have reframed our argument to emphasize that directly constraining standard deviation (continuous) or logit range (discrete) offers a straightforward and interpretable mechanism for controlling exploration coverage.
>
> 3. "[entropy coefficient schedules] are difficult to design and sensitive to reward scaling"
>
> We would like to further distinguish between manual entropy coefficient tuning and the broader category of entropy scheduling. We agree that the paper (Abdolmaleki et al. 2015) uses an entropy schedule that is insensitive to reward scaling. Yet, an entropy coefficient is sensitive to reward scaling, as its effective strength depends on the reward magnitude (Soft Actor-Critic Algorithms and Applications, Haarnoja et al., 2018b). We have revised Section 1 to clarify that our work specifically focuses on comparing against fixed or heuristically annealed entropy coefficients in complex tasks.
>
> 4.  "directly constraining  is more intuitive and numerically stable [compared to entropy]"
>
> We agree that the unbounded nature of Gaussian entropy is not, by itself, sufficient evidence to claim a relative lack of numerical stability. Accordingly, we have removed this claim from Section 2.1 and Section 3.2 to ensure a more objective presentation of the approaches.
>
> 5. "entropy acts as a lossy summary statistic: multiple distributions can share identical entropy values while having vastly different minimum action probabilities"
>
> We have removed the direct comparison between entropy and logit ranges as "lossy" statistics in Section 2.1 to avoid an over-generalized claim. Instead, we have reframed the discussion to focus on the beneficial feature of logit range constraints: specifically, the ability to provide an analytical lower bound on the probability of all actions, which prevents the irreversible loss of potentially optimal behaviors.
>
>
> ## Terminology
>
> 1. Flatness is not precisely defined but introduced per example
>
> We have restructured the opening of Section 3.2 to explicitly clarify that "flatness" is an informal term for action space coverage, not a precise mathematical definition. The revised text now reads: "for each action space type, we select a scalar quantity $F(\pi)$ that captures action space coverage. We refer to this informally as the flatness of the policy." Each per-space instantiation is now explicitly motivated by its correlation with coverage.
>
> 2. Lower bounds should be stated as heuristically chosen
>
> We have revised the presentation of the lower bounds. The new paragraph under Section 3.2 now states: "As a simple heuristic, we choose f to be an affine function of the goal achievement." The per-space paragraphs further emphasize this: for continuous actions, "the corresponding lower bound $\sigma_{\rm LB}$ (i.e., $F_{\rm LB}$ for the continuous case) is heuristically set as" (before Eq. 2); for discrete actions, "following the same heuristic as in continuous space, the corresponding lower bound $\delta_{z,\text{LB}}$ (i.e., $F_\text{LB}$ for the discrete case) is set as" (before Eq. 5).
>
> We have also taken care to ensure a more objective presentation throughout the manuscript, as you recommended. We hope the revised camera-ready version addresses all concerns satisfactorily. We are grateful for the constructive feedback, which has substantially improved the clarity of the paper.
>
> Best regards,
>
> The Authors